# Multiple redox switches of the SARS-CoV-2 main protease in vitro provide opportunities for drug design

Lisa-Marie Funk[1,2], Gereon Poschmann [3], Fabian Rabe von Pappenheim [1,2], Ashwin Chari [4], Kim M. Stegmann[5], Antje Dickmanns[5], Marie Wensien[1,2], Nora Eulig [1,2], Elham Paknia[4], Gabi Heyne[4], Elke Penka[1,2], Arwen R. Pearson [6], Carsten Berndt[7], Tobias Fritz [8], Sophia Bazzi[8], Jon Uranga[8], Ricardo A. Mata [8], Matthias Dobbelstein[5], Rolf Hilgenfeld[9,10], Ute Curth[11] & Kai Tittmann [1,2] ✉

Besides vaccines, the development of antiviral drugs targeting SARS-CoV-2 is critical for preventing future COVID outbreaks. The SARS-CoV-2 main protease (M^pro), a cysteine protease with essential functions in viral replication, has been validated as an effective drug target. Here, we show that M^pro is subject to redox regulation in vitro and reversibly switches between the enzymatically active dimer and the functionally dormant monomer through redox modifications of cysteine residues. These include a disulfide-dithiol switch between the catalytic cysteine C145 and cysteine C117, and generation of an allosteric cysteine-lysine-cysteine SONOS bridge that is required for structural stability under oxidative stress conditions, such as those exerted by the innate immune system. We identify homo- and heterobifunctional reagents that mimic the redox switching and inhibit M^pro activity. The discovered redox switches are conserved in main proteases from other coronaviruses, e.g. MERS-CoV and SARS-CoV, indicating their potential as common druggable sites.

The COVID-19 pandemic, caused by the severe acute respiratory syndrome coronavirus 2 (SARS-CoV-2), has constituted the largest global health crisis in the recent past with over six million deaths and over 768 million confirmed cases worldwide[1]. Although the development of vaccines has been instrumental in the reduction of severe progression and lethality of the disease, antiviral drugs are required to complement vaccination in high-risk groups, and for controlling sudden future outbreaks[2,3]. Also, the genetic diversity and rapid evolution of SARS-CoV-2 has led to the emergence of virus variants, for which vaccination has reduced efficiency[4–6]. As SARS-CoV-2 or related viruses are expected to remain a global threat in the future, the development of antiviral drugs becomes increasingly important.

[1]Department of Molecular Enzymology, Göttingen Center of Molecular Biosciences, Georg-August University Göttingen, Julia-Lermontowa-Weg 3, D-37077 Göttingen, Germany. [2]Max-Planck-Institute for Multidisciplinary Sciences, Am Fassberg 11, D-37077 Göttingen, Germany. [3]Institute of Molecular Medicine, Proteome Research, Medical Faculty and University Hospital Düsseldorf, Heinrich-Heine University Düsseldorf, Universitätsstraße 1, 40225 Düsseldorf, Germany. [4]Department of Structural Dynamics, Max-Planck-Institute for Multidisciplinary Sciences, Am Fassberg 11, D-37077 Göttingen, Germany. [5]Institute of Molecular Oncology, University Medical Center Göttingen, Justus-von-Liebig-Weg 11, 37077 Göttingen, Germany. [6]Institute for Nanostructure and Solid-State Physics, Hamburg Centre for Ultrafast Imaging, Hamburg University, HARBOR, Luruper Chaussee 149, Hamburg 22761, Germany. [7]Department of Neurology, Medical Faculty, Heinrich-Heine University Düsseldorf, Moorenstr. 5, 40225 Düsseldorf, Germany. [8]Institute of Physical Chemistry, Georg-August University Göttingen, Tammannstraße 6, D-37077 Göttingen, Germany. [9]Institute for Biochemistry, Lübeck University, Ratzeburger Allee 160, 23562 Lübeck, Germany. [10]German Center for Infection Research, Hamburg - Lübeck-Borstel-Riems Site, University of Lübeck, Ratzeburger Allee 160, 23562 Lübeck, Germany. [11]Institute for Biophysical Chemistry, Hannover Medical School, Carl-Neuberg-Straße 1, 30625 Hannover, Germany. ✉e-mail: ktittma@gwdg.de

Major therapeutic strategies for the treatment of COVID-19 include the application of neutralizing antibodies/nanobodies and small-molecule drugs targeting vital enzymes of the viral replication machinery[7–11]. In the latter context, the SARS-CoV-2 main protease M[pro] is a particularly promising drug target[12–14]. It proteolytically processes the viral polyproteins pp1a and pp1ab at no less than 11 cleavage sites, and thereby also ensures its own release. Its biological function in the viral replication cycle, along with the absence of a closely related human homologue, establishes M[pro] as a propitious drug target. The structure determination of SARS-CoV-2 M[pro] sparked the development of several classes of inhibitors that bind either to the active site and covalently modify the catalytic cysteine or to allosteric sites[12–22]. These efforts culminated in the design of Paxlovid™ (Pfizer), an orally administered FDA-approved

antiviral drug which contains nirmatrelvir, an inhibitor targeting SARS-CoV-2 M[pro] [23].

M[pro] is a cysteine protease that contains a catalytic dyad consisting of the nucleophilic cysteine 145 (C145) and histidine 41 (H41)[12]. In total, M[pro] contains 12 cysteine residues per chain (306 residues) (Fig. 1a, Supplementary Fig. 1), which amounts to ~4% cysteines. This is a statistically unusual high cysteine abundance for a viral protein[24]. The involvement of catalytic cysteine residues is a potential Achilles heel for viral replication, as oxidative stress exerted by the host innate immune system in response to viral infection may irreversibly (over) oxidize the cysteines and thus inactivate the enzyme and block replication[25,26]. Although M[pro] resides in the cytoplasm, which is typically considered to be of reducing nature, it has been established that oxidative bursts or even physiological redox signaling based on

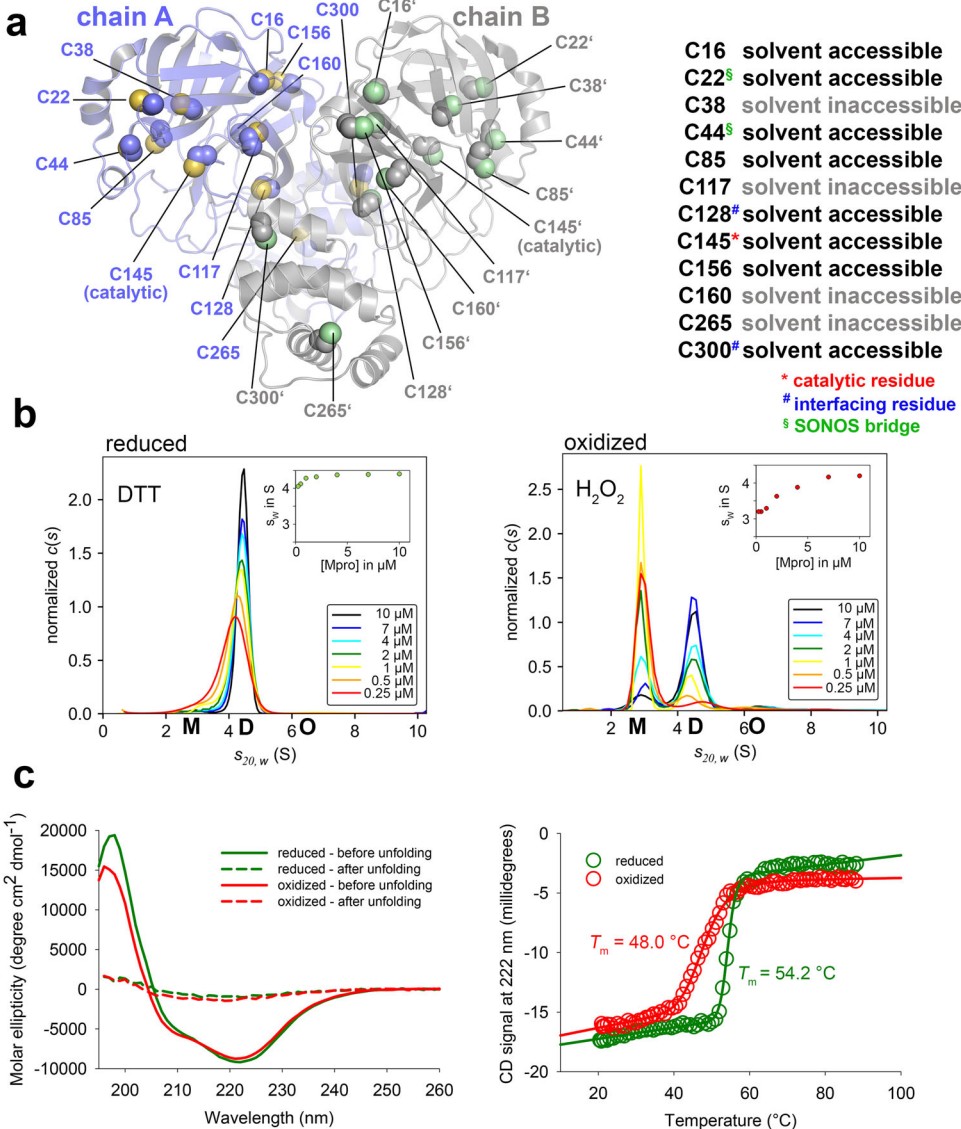

**Fig. 1 | Structure and redox properties of SARS-CoV-2 main protease (M[pro]).**
**a** Structure of the M[pro] dimer (pdb code 7KPH) highlighting the positions, structural properties and functions of cysteine residues. The two monomers of the functional dimer and corresponding cysteines are colored individually. A close-up of the active site and proximal cysteines is shown in Supplementary Fig. 1. **b** Sedimentation velocity analysis of SARS-CoV-2 M[pro] in a concentration range from 0.25 to 10 µM under either reducing (left panel, 1 mM DTT) or oxidizing (right panel, 1 mM H2O2) conditions indicate a redox-dependent monomer ⇔ dimer equilibrium with apparent equilibrium constants of $K_{app}$ <0.25 µM for the reduced enzyme and of

about 2.5 µM for the oxidized enzyme. Insets show $s_w$ binding isotherms, as calculated from the corresponding c(s) distributions. Abbreviations: M, monomer ($s_{20,w}$ = 2.9 S); D, dimer ($s_{20,w}$ = 4.5 S); O, oligomers (($s_{20,w}$ = 6.3 S). **c** Secondary structure (left panel) and thermal unfolding (right panel) analysis of M[pro] by far-UV CD spectroscopy under reducing and oxidizing conditions. Note the slightly reduced helical content (lower signal at 222 nm) and the decreased melting temperature of the oxidized enzyme ($T_m$ = 48.0 °C) versus the reduced counterpart ($T_m$ = 54.2 °C). Further note the decreased cooperativity of unfolding (decreased steepness of transition) of the oxidized enzyme.

enzymatic production of reactive oxygen species (ROS) such as $H_2O_2$ leads to locally oxidizing conditions and subsequent oxidation of protein thiols in the cytosol[27]. We recently reported the discovery of lysine-cysteine redox switches in proteins consisting of NOS (nitrogen-oxygen-sulfur) and SONOS (sulfur-oxygen-nitrogen-oxygen-sulfur) bridges[28,29]. Interestingly, $M^{pro}$ is amongst this class of proteins suggesting the possibility that it is redox regulated[29]. Specifically, an allosteric SONOS bridge consisting of two cysteines (C22, C44) and one lysine (K61) within one protein chain was detected by our mining of the protein data base and independent structural studies of Liu and coworkers (Supplementary Fig. 2)[29,30]. C44 is close in sequence to catalytic residue H41 and located on a flexible loop (residues 44-53). It points either towards the active site and contacts Y54 ("in" conformation) or towards the protein surface ("out" conformation). The loop is structurally highly flexible as revealed by our molecular dynamics (MD) simulations (Supplementary Fig. 3) and also reported by temperature-dependent structure analysis of $M^{pro}$ [31]. Neutron crystallographic studies on $M^{pro}$ showed that both the catalytic C145 as well as SONOS residues C22 and C44 exist as the deprotonated thiolate, which would facilitate their oxidation[32]. Independent structural studies have indicated that the catalytic cysteine C145 of $M^{pro}$ is susceptible to oxidation and forms various oxidation products including mono-oxidized (sulfenic acid) and di-oxidized (sulfinic acid) species (Supplementary Fig. 4), which rapidly interconvert to the tri-oxidized (sulfonic acid) form. The latter two forms are considered to be irreversible modifications and would lead to a dysfunctional enzyme and an arrest of viral replication[33]. In case of the closely related $M^{pro}$ from SARS-CoV, a disulfide modification between the catalytic C145 and C117 had been reported for a variant, in which residue N28 had been replaced[34].

In this work, we demonstrate that the SARS-CoV-2 $M^{pro}$ is subject to redox regulation in vitro and contains multiple redox switches including a disulfide-dithiol switch between the catalytic residue C145 and C117, as well as a SONOS switch of residues C22, C44 and K61. These redox switches prevent $M^{pro}$ from becoming irreversibly overoxidized and structurally destabilized under oxidizing conditions.

## Results and Discussion
### Redox-regulated enzymatic activity and oligomeric equilibria of $M^{pro}$

We had initially observed that $M^{pro}$ loses enzymatic activity over a couple of days on ice when kept in non-reducing buffer (aerated buffer devoid of reductants such as e.g. DTT). Enzymatic activity could be fully restored when the enzyme was reacted with DTT. In order to test for and analyze a potential redox regulation of $M^{pro}$ quantitatively, we subjected the protein to different levels of oxidative insult with $H_2O_2$ including a) 100 μM $H_2O_2$ as an upper limit for physiologically relevant oxidative stress conditions, b) 1 mM $H_2O_2$ or c) 20 mM $H_2O_2$ as a supraphysiological concentration[35,36].

We first measured the enzymatic activity under reducing versus oxidizing conditions and tested for reversibility of redox switching. The data are exemplary shown for the treatment with 100 μM $H_2O_2$ (Fig. 2). We observed a progressive but essentially reversible loss of enzymatic activity over time that could be fully reversed upon treatment with the reductant DTT. When kept on ice, inactivation takes place over a time of 10–20 hours. This would be seemingly physiologically relevant in view of the SARS-CoV-2 replication time and reported eclipse period of 10 h at 37–40 °C[37] as the rate of chemical reactions is typically 2-3fold higher per 10 K temperature increase. Using this approximation, the oxidation of $M^{pro}$ by 100 μM $H_2O_2$ should be 8–12 time faster at 37–40 °C compared to 0 °C. The inactivation upon treatment with 1 mM $H_2O_2$ proceeds – as expected – faster, that is on a time scale of a few hours and is also fully reversible (Supplementary Fig. 5a and Table 1). At 20 mM $H_2O_2$, however, enzymatic activity is irreversibly lost (Supplementary Fig. 5a). This observation

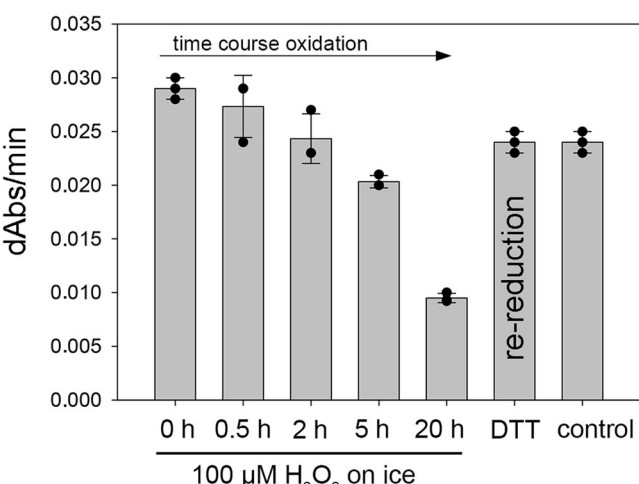

**Fig. 2 | Redox dependence of enzymatic activity of SARS-CoV-2 $M^{pro}$.** Enzymatic activity of $M^{pro}$ after incubation with 100 μM $H_2O_2$ on ice for different time points (assay conditions are detailed in the Methods section). Note the progressive loss of enzymatic activity over time. Re-reduction of the oxidized enzyme (20 h reaction time) with DTT overnight on ice fully restores enzymatic activity as compared to an enzyme control sample, which was kept on ice for 20 h without adding oxidizing or reducing agents. All measurements were carried out in triplicate ($n = 3$) and are shown as mean ± s.d. Almost identical results were obtained in two independent biological replicates. Source data are provided as a Source Data file.

indicates that the protein becomes irreversibly overoxidized under these conditions, presumably through oxidation of the catalytic Cys145. At concentrations up to 1 mM $H_2O_2$, the catalytic cysteine Cys145 is well protected against overoxidation involving either sulfenic acid, a disulfide or a lysine-cysteine switch.

We further conducted gel filtration experiments with the reduced and oxidized protein (Supplementary Fig. 5b). For practical reasons, we opted to use 1 mM $H_2O_2$ in case of oxidizing conditions as a) the oxidized protein can be obtained within a few hours at these $H_2O_2$ concentrations (see above) and b) switching is fully reversible. In the reduced state, $M^{pro}$ is almost exclusively present as the dimer under the conditions used (only the dimer has enzymatic activity)[12]. When treated with 1 mM $H_2O_2$, however, the $M^{pro}$ dimer undergoes a dissociation leading to the formation of a marked monomer fraction. A re-reduction of oxidized $M^{pro}$ leads to the quantitative formation of the dimer indicating a fully reversible redox switch on the oligomer level. Interestingly, when using supraphysiological concentrations of $H_2O_2$–that is 20 mM – monomerization is not observed.

To quantitatively assess the oligomeric equilibrium between monomer and dimer under reducing and oxidizing conditions, we performed analytic ultracentrifugation experiments (Fig. 1b, Supplementary Figs. 6 & 7). Under reducing conditions, the $M^{pro}$ dimer hardly dissociates even at the lowest protein concentration tested (0.25 μM) suggesting a $K_D^{app}$ of <0.25 μM. In contrast, under oxidizing conditions––either in the presence of $H_2O_2$ (0.1 mM, 1 mM) or, alternatively, $O_2$ in a non-reducing buffer––$M^{pro}$ clearly exists in a monomer-dimer equilibrium. From the transition range of the $s_w$ binding isotherms, an apparent $K_D$ of about 2.5 μM could be estimated, as reported before[12]. However, an exact determination is prevented by the inability to fit the isotherms to a monomer:dimer two-state model. That notwithstanding the data clearly indicate that the equilibrium constant is at least one order of magnitude larger compared to the reduced protein. The shift of the oligomeric equilibrium under changing redox conditions is fully reversible for

## Table 1 | Redox-dependent enzymatic activities of SARS-CoV-2 M^pro wild-type and variants

| M^pro Variants | Reduced [a] Relative activity (%) | Oxidized [b] Relative activity (%) | Reactivation (after 200s) [c] | |
|---|---|---|---|---|
| | | | Relative activity (%) | Change (%) |
| Wild-type | 100.0 ± 2.6[d] | 30.5 ± 1.0 | 34.0 ± 1.0 | +3.5% |
| | | | Fully reversible overnight[e] | |
| **Cys-Ser variants** | | | | |
| C16S | 75.3 ± 1.1 | 25.4 ± 0.1 | 27.1 ± 0.2 | +1.7% |
| C22S | 84.4 ± 1.2 | 26.3 ± 0.2 | 31.2 ± 0.8 | +4.9% |
| C38S | 92.1 ± 0.5 | 31.8 ± 0.3 | 36.4 ± 0.4 | +4.6% |
| C44S | 16.4 ± 0.1 | 7.3 ± 0.3 | 7.6 ± 0.4 | +0.3% |
| C85S | 71.2 ± 1.7 | 26.4 ± 0.8 | 29.5 ± 0.1 | +3.1% |
| C117S | 41.9 ± 0.6 | 16.7 ± 0.1 | 14.2 ± 1.1 | −2.5% |
| | | | irreversible overnight[e] | |
| C128S | 74.3 ± 0.6 | 22.5 ± 0.6 | 28.4 ± 1.1 | +5.9% |
| C145S | n. a.[f] | n. a. | n. a. | |
| C156S | 77.5 ± 1.8 | 28.7 ± 0.7 | 35.0 ± 0.9 | +6.3% |
| C160S | 92.7 ± 3.0 | 28.2 ± 0.5 | 30.1 ± 0.1 | +1.9% |
| C265S | 85.1 ± 2.3 | 30.7 ± 0.4 | 34.4 ± 1.9 | +3.7% |
| C300S | 92.7 ± 0.3 | 32.1 ± 0.4 | 35.9 ± 2.0 | +3.8% |
| **Additional SONOS variants** | | | | |
| C44A | 29.4 ± 0.4 | 12.4 ± 0.5 | 11.1 ± 0.4 | −1.3 |
| C22S_C44S | 10.4 ± 0.4 | 4.0 ± 0.0 | 2.3 ± 0.2 | −1.7 |
| K61A | 55.2 ± 2.0 | 20.2 ± 2.4 | 18.7 ± 2.1 | −1.5 |
| K61A_C22S | 42.4 ± 1.9 | 16.6 ± 2.2 | 17.2 ± 0.7 | +0.6 |
| K61A_C44S | 10.2 ± 0.3 | 3.2 ± 0.0 | 3.5 ± 0.1 | +0.3 |
| K61A_C22S_C44S | 7.0 ± 1.4 | 3.0 ± 0.1 | 2.0 ± 0.3 | −1.0 |
| | | | Partially reversible overnight[e] | |
| Y54F | 19.4 ± 0.0 | 7.1 ± 1.8 | 8.2 ± 0.5 | +1.1 |

[a]in presence of 1 mM DTT.

[b]after oxidation with 1 mM H₂O₂ for 2 h on ice.

[c]re-activation was initiated by addition of 20 mM DTT to the oxidized enzyme and kinetically analyzed 200 s after reduction.

[d]100% activity refers to the activity of wild-type M^pro under reducing conditions.

[e]see Supplementary Fig. 5 (wild-type), Supplementary Fig. 11 (variant C117S) and Supplementary Fig. 12 (variant K61A_C22S_C44S).

[f]no measurable activity.

Experimental details are provided in the Supplementary Methods.

treatment with 100 μM or 1 mM $H_2O_2$ as well with nonreducing aerated buffer (Supplementary Figs. 6 & 7). The dissociation of a protein oligomer under oxidizing conditions is intriguing, typically, the reverse effect is observed for redox-sensitive proteins where reduction of interchain disulfide bridges leads to deoligomerization[38]. Structure analysis of M^pro by far-UV circular dichroism (CD) spectroscopy under oxidizing and reducing conditions indicates small but reproducible structural differences between the two states based on secondary structure content (Fig. 1c). Upon oxidation, the fraction of α-helices slightly decreases (lower signal at 222 nm), while the fraction of ß-strands increases (higher signal at 210 nm). Also, the oxidized and reduced protein exhibit different thermal stabilities and different cooperativities of unfolding. The melting temperature of oxidized M^pro (Tm = 48.0 °C) is ~6 °C lower than that of the reduced enzyme (Tm = 54.2°C). The transition from the folded to the unfolded state is less steep in case of the oxidized protein indicating a reduced cooperativity of unfolding.

## Identification of M^pro cysteines underlying redox switching

In order to identify cysteine residues that are part of the redox switch(es), we generated single-site variants with cysteine-to-serine substitutions for all 12 cysteines. In addition, we produced single, double and triple mutants with individual and combined exchanges of the SONOS bridge residues C22, C44 and K61 including residue Y54 that directly interacts with C44. First, we analyzed the enzymatic activity of all variants under reducing and oxidizing conditions and tested whether a putative redox-induced change in activity is reversible (Fig. 3a, Table 1). For practical reasons, we used 1 mM $H_2O_2$ for oxidizing conditions as wild-type M^pro undergoes fully reversible redox switching under these conditions and oxidized protein is obtained in a couple of hours. As expected, variant C145S, in which the catalytic cysteine has been replaced, exhibits no measurable enzymatic activity. Most cysteine variants are almost as active as the wild-type protein with variants C117S and C44S being notable exceptions. Variant C44S exhibits the lowest residual activity (16%), while variant C117S is slightly more active (42% residual activity) (Table 1). The markedly reduced activity of variant C44S comes not unexpected as residue C44 is very close to catalytic residue H41 (see Supplementary Fig. 2). A full kinetic Michealis-Menten analysis indicates that the reduction in activity of variant C44S is due to a decreased catalytic constant ($k_{cat}$) rather than impaired substrate binding (Supplementary Fig. 8). Both wild-type M^pro as well as variant C44S exhibit positive cooperativity with a Hill coefficient of $n_H$>1. Double and triple variants with multiple exchanges of SONOS residues lead to enzyme variants with almost abolished enzymatic activity. Interestingly, variant Y54F, in which the tyrosine that interacts with SONOS residue C44 is replaced, shows a similar catalytic deficiency (19%) as variant C44S suggesting that both residues are required for full catalytic competence of the active site. To define the structural basis of the markedly reduced enzymatic activity in variants C44S and Y54F, we crystallized both variants and compared the atomic structure with the known structure of the wild-type enzyme (X-ray statistics in Supplementary Table 1)[39]. In the case of variant C44S, we were able to obtain a structural snapshot of the covalent acyl intermediate formed with the C-terminal glutamine of a symmetry-related M^pro molecule as previously reported for wild-type M^pro (Supplementary Fig. 9). This structural analysis reveals in both cases small structural changes mostly confined to the active site, notably of residue H41 that forms the catalytic dyad with C145 (Supplementary Fig. 9, Supplementary Fig. 10). This is not surprising given that C44 and H41 are close in sequence as discussed before.

We then measured the enzymatic activity under defined oxidizing conditions and tested whether a putative loss of activity is reversible. For the latter, we used the early onset of reactivation over the first 200 s after re-reduction with DTT as a proxy and analyzed variants with a kinetic phenotype in more detail with reduction taking place overnight. Treatment of M^pro wild-type and variants with 1 mM $H_2O_2$ for 2 h on ice results in decreased enzymatic activities to a similar extent in all proteins (~3-fold reduction) pinpointing the central role of catalytic residue C145 as a major site of redox modification (the only residue present in all tested variants) (Table 1). Re-reduction of the protein with DTT leads to a reactivation of enzymatic activity in all cysteine variants akin to the wild-type enzyme with the notable exception of variant C117S (Fig. 3a, Table 1). The activity of this variant is irreversibly lost upon oxidation even when reduction with DTT took place over night (Supplementary Fig. 11). This is a clear indication that C117 is part of a redox switch involving C145, most likely in the form of a disulfide-dithiol switch. Interestingly, a C117-C145 disulfide was reported for a SARS-CoV M^pro variant but not the wild-type protein[34]. In variant C117S, where no C145-C117 disulfide can be formed, C145 might not be protected against overoxidation and thus explain the irreversible nature of redox switching. For some of the SONOS variants such as triple variant C22S_C44S_K61A, only a partial recovery of enzymatic activity can be observed but only after long incubation times with reductant

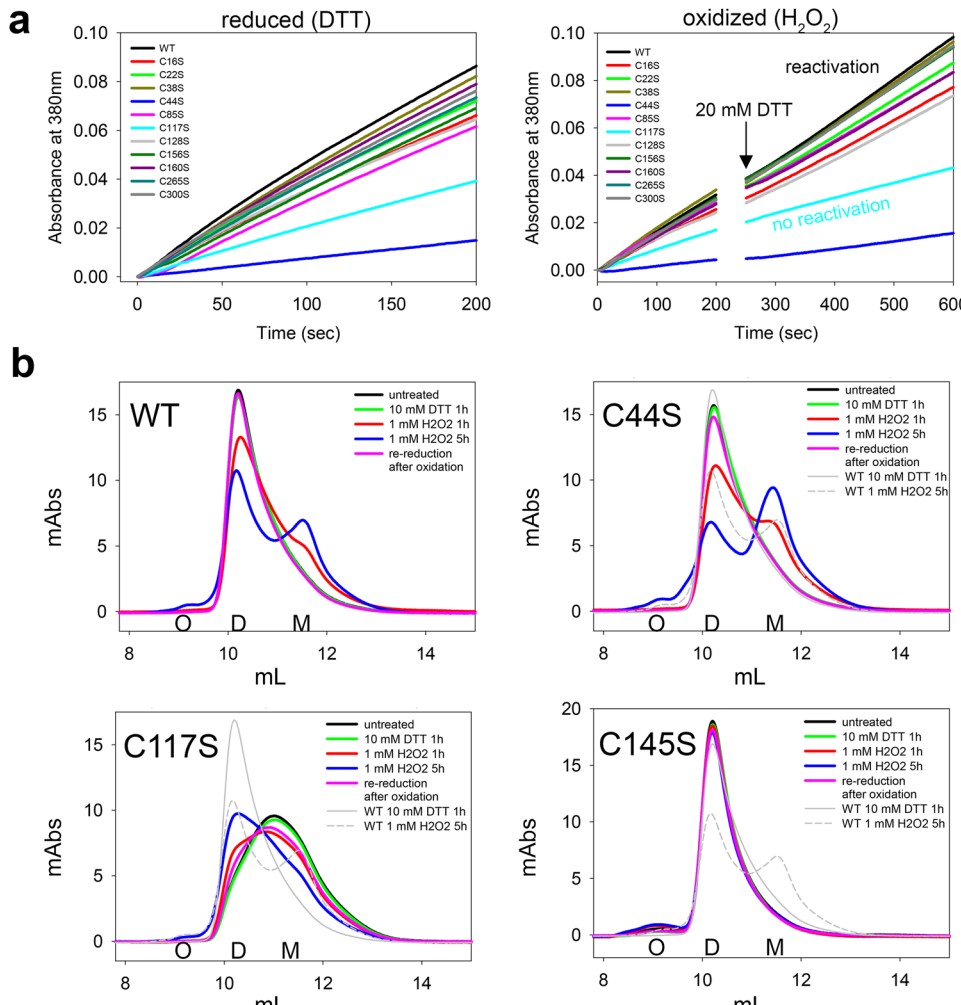

**Fig. 3 | Redox-dependent enzymatic activity and oligomeric state of SARS-CoV-2 main protease (M$^{pro}$) wild-type (WT) and cysteine variants. a** Progress curves of substrate turnover for the reduced (left panel, 1 mM DTT) versus oxidized (right panel, 2 h 1 mM H$_2$O$_2$) enzyme. The relative activities are summarized in Table 1. In case of the oxidized enzyme, 20 mM DTT were added after a reaction time of 200 s to reactivate the enzyme by re-reduction. Reactivation was monitored up to a total reaction time of 10 min. Note that all enzyme variants except for C117S become reactivated. Variant C145S with a substitution of the catalytic cysteine is enzymatically inactive and not shown. All experiments were done in duplicate and with two independent biological replicates. **b** Gel filtration analysis of the oligomeric state of M$^{pro}$ wild-type (WT) and selected, phenotypically outstanding cysteine variants in the reduced state and after different reaction times with H$_2$O$_2$. Abbreviations: O, oligomer; D, dimer; M, monomer. Note the progressive formation of the monomer with increasing oxidation times in case of the WT enzyme. For variant C44S, a larger fraction of the monomer is observed that likely reflects a kinetic rather than a thermodynamic effect (Supplementary Fig. 13). Variant C117S is phenotypically unique in the stabilization of the monomer under reducing conditions and formation of the dimer upon oxidation. In contrast to the WT and all other cysteine variants tested, the redox switch on the quaternary level is not fully reversible for C117S. Variant C145S does not undergo monomerization in the course of oxidation under the conditions used highlighting the essential role of C145 for the redox switch.

showcasing the structural importance of the SONOS motif for the correct functioning of the redox switch(es) of M$^{pro}$ (Table 1, Supplementary Fig. 12).

The analysis of the monomer-dimer equilibrium by gel filtration experiments under reducing vs. oxidizing (1 mM H$_2$O$_2$) conditions further substantiated the critical roles of potential disulfide-forming residues (C145, C117) and the SONOS residues for the redox switching of M$^{pro}$ (Fig. 3b, Supplementary Tables 2 & 3, Supplementary Data 1). Variant C145S was the only variant, for which no marked monomer formation was detectable under the conditions used. Variant C117S was unique in forming a detectable fraction of the monomer already under reducing conditions. Intriguingly, upon oxidation, the oligomeric equilibrium shifted to the dimer as opposed to the wild-type protein that shifts to the monomer. This indicates that the mutation of C117 leads to a structural rearrangement that is not confined to the local environment of C117 but also changes the structure of the dimer interface. Notably, the redox-dependent shift on the quaternary level is

not reversible for variant C117S, while all other variants tested undergo a fully reversible switching. This observation suggests that oxidation of C117S leads to an irreversibly modified protein, presumably with an overoxidized catalytic C145. For SONOS variant C44S, we observed a somewhat larger fraction of the monomer that is likely to result from a faster oxidation reaction as the analytical ultracentrifugation experiments indicate similar dissociation constants for the variant under reducing and oxidizing conditions as for M$^{pro}$ wild-type (Supplementary Fig. 13). While the analytical ultracentrifugation experiments for variant C117S are compatible with the gel filtration analysis, variant C145S undergoes dissociation upon oxidation in the AUC experiments, albeit only at low concentrations (Supplementary Fig. 14). This might indicate that the monomer-dimer equilibrium in this variant is not solely thermodynamically controlled but also kinetically (gel filtration experiments are conducted immediately after oxidative insult that is within 30 min, ultracentrifugation analysis is preceded by an overnight incubation to allow the system to equilibrate). In general, the gel

filtration experiments indicate that all variants, particularly these with substitutions of SONOS residues, tend to form larger fractions of higher oligomers/aggregates under oxidizing conditions suggesting a role of the SONOS bridge for structural stabilization (Supplementary Table 3).

The comparative analysis of all proteins by far-UV CD spectroscopy regarding secondary structure content and thermal unfolding (melting temperature and cooperativity of unfolding) identified "disulfide variants" C145S and C117S as phenotypically conspicuous (Supplementary Fig. 15, Supplementary Data 2). In contrast to the wild-type enzyme, the slight increase of ß-strand elements at the expense of α-helical content upon oxidation is not observed for these variants (Supplementary Table 4). Also, the melting temperatures of the reduced and oxidized counterparts are almost identical (Supplementary Fig. 15, Supplementary Table 5, Supplementary Data 3). This contrasts our observations with wild-type $M^{pro}$, where the reduced and oxidized forms exhibited different melting temperatures and cooperativities of unfolding (see Fig. 1). SONOS variants, in particular those containing an exchange of residue K61, are very susceptible to aggregation under oxidizing conditions and exhibit an atypical early onset of thermal denaturation (30–35 °C) with almost no cooperativity of unfolding, indicating a very loosely structured protein (Supplementary Fig. 15c). Under reducing conditions, the protein is structurally stable. This would imply a structurally stabilizing function of the SONOS bridge under oxidizing conditions. The X-ray crystallographic analysis of the K61A variant in complex with the acyl intermediate formed between C145 and Q306 of a symmetry-related $M^{pro}$ molecule indeed reveals marked structural changes throughout the whole molecule with an r.m.s.d. of the Cα-carbons of 2.87 Å for chain A and 3.10 Å for chain B, respectively, compared to the wild-type structure (Supplementary Fig. 16).

The structural changes for K61A are clearly more pronounced than observed for variants C44S (0.89/0.91 Å) and Y54F (0.24 Å), highlighting the structural importance of K61.

### An intramolecular disulfide and a SONOS bridge as key redox switch elements of $M^{pro}$

We next set out to identify the redox modifications of $M^{pro}$ by mass spectrometry-based redox proteomics and Western blot analysis (Fig. 4). This included the analysis of oxidized cysteine species (sulfenic acid (mono-oxidized) and sulfonic acid (tri-oxidized)), disulfide bridges and lysine-cysteine NOS/SONOS bridges. Determination of sulfenic acids via dimedone tagging confirmed that the cysteines in $M^{pro}$ exhibit in general a rather low sensitivity towards oxidation (Fig. 4a). Only treatment with supraphysiological concentrations of $H_2O_2$ (20 mM) leads to a substantial formation of sulfenic acids and indicates formation of irreversible overoxidation (sulfenic acid easily oxidizes to higher oxidation states). At $H_2O_2$ concentrations up to 1 mM, mass spectrometry in combination with dimedone tagging identified residues C145 (catalytic residue), C156, and C300 as becoming sulfenylated (Fig. 4b, Supplementary Fig. 17). A $H_2O_2$ concentration-dependent sulfonylation was found for the catalytic cysteine C145 and, to a lesser extent, also for C117. Both cysteines form relatively small fractions of the irreversibly oxidized sulfonic acid up to 1 mM $H_2O_2$ in line with the reversible redox switching of $M^{pro}$ under these conditions (see above). Residues C85 and C300 are found to be oxidized to sulfonic acid particularly at supraphysiological concentrations (20 mM), whereas no $H_2O_2$ concentration-dependent oxidation of the other cysteines was obvious (Supplementary Figs. 18, 19).

The existence of SONOS-linked peptides (C22-K61-C44) could not be directly proven by mass spectrometry of the proteolytically digested protein similar to the initially discovered NOS crosslink in a transaldolase[28], but we noticed that residues C22 and C44 were only accessible for alkylation after reduction (following an initial oxidation with $H_2O_2$) implicating a previous oxidized state of both sites. As C22

and C44 were neither found to be sulfenylated, sulfinylated, sulfonylated (traces of sulfonylated C22 were found at supraphysiological $H_2O_2$ concentrations) nor in a disulfide linkage, this might be considered as indirect evidence for the existence of the SONOS bridge protecting those residues from getting further oxidized at physiologically relevant $H_2O_2$ concentrations. Mass spectrometric evidence for the existence of the SONOS bridge was recently provided by Liu and colleagues who analyzed the undigested protein[30].

Three disulfide-linked cysteine pairs, C117/C145, C117/C300, and C145/C300 were identified repetitively in peptides pairs of $M^{pro}$ after oxidation with 100 μM and 1 mM $H_2O_2$ (Fig. 4b, Supplementary Fig. 20). Since the occurrence of C117/C300 and C145/C300 disulfides at physiologically relevant $H_2O_2$ concentrations was in the range of the DTT-treated $M^{pro}$, we conclude that the C145/C117 disulfide is a decisive modification underlying the redox switching of $M^{pro}$ rather than C117/C300 or C145/C300. This is also supported by our mutagenesis studies, in which C117S was the only variant that exhibited an irreversible oxidation based on enzymatic activity and the oligomeric equilibrium (see above). Also, disulfides between C300 and either C145 or C117 would be formed between the two chains of the functional dimer (see Supplementary Fig. 1) and thus not be compatible with the detected dimer dissociation under oxidizing conditions. Inspection of the X-ray structure of $M^{pro}$ determined in the reduced state indicates that residues C145 and C117 cannot directly form a disulfide bond as residue N28 is bound in between the two side chains (Fig. 4c). Interestingly, N28 was previously demonstrated to be important for catalysis and dimer stability in $M^{pro}$ of SARS-CoV. Substitution of N28 by alanine led to a variant with markedly decreased enzymatic activity and dimer stability[34]. In the variant, catalytic C145 and C117 were found to form a disulfide linkage akin to our findings for wild-type $M^{pro}$ from SARS-CoV-2.

To obtain insights into why variant C117S, in which the C117-C145 disulfide-dithiol switch has been defunctionalized, is not reversibly switching as the wild-type protein, we conducted a head-to-head redox proteomics analysis of C117S versus wild-type $M^{pro}$. First, we analyzed the oxidation states of catalytic residue C145 under different oxidizing conditions (Supplementary Fig. 21). As expected, overoxidation (sulfonylation) of catalytic C145 in C117S is increased, but not to an extent that can fully explain the irreversible nature of redox switching in the variant. We therefore analyzed other modifications including disulfide bridges (Supplementary Fig. 22). As discussed above, wild-type $M^{pro}$ forms the C117-C145 disulfide as the major linkage. In C117S, however, a marked increase of the C145-C300 disulfide linkage is observed relative to the wild-type. Owing to the relatively close spatial proximity of C300 of one chain of the $M^{pro}$ dimer to the catalytic site of the neighboring chain (see Supplementary Fig. 1), this disulfide bridge is very likely an interchain crosslink. This finding would explain, why oxidation of variant C117S leads to formation of the dimer (see Fig. 3b). The irreversible nature of the redox switching in C117S would require that the formed C145-C300 disulfide is shielded from the solvent such that reductants as DTT cannot directly react thus constituting a kinetically stable disulfide under the conditions used. Overall, variant C117S is slightly more susceptible to oxidation at rather low $H_2O_2$ concentrations of 20 and 100 μM than the wild-type protein using the Western blot analysis of sulfenylation as a readout (Supplementary Fig. 23).

In the absence of an experimental structure of SARS-CoV-2 $M^{pro}$ with a C145-C117 disulfide link, we conducted MD simulations for $M^{pro}$ with and without the disulfide linkage between C145 and C117 as recently reported for inhibitor profiling of $M^{pro}$ [40]. One of the main structural differences is--as expected--the displacement of residue N28 (Fig. 5, Supplementary Fig. 24). Computed dimerization enthalpies show that the formation of the disulfide link reduces the latter by 4.2 kcal/mol (Fig. 5). We thus hypothesize that redox switching of Mpro entails a structural reorganization upon oxidation that brings C145 and

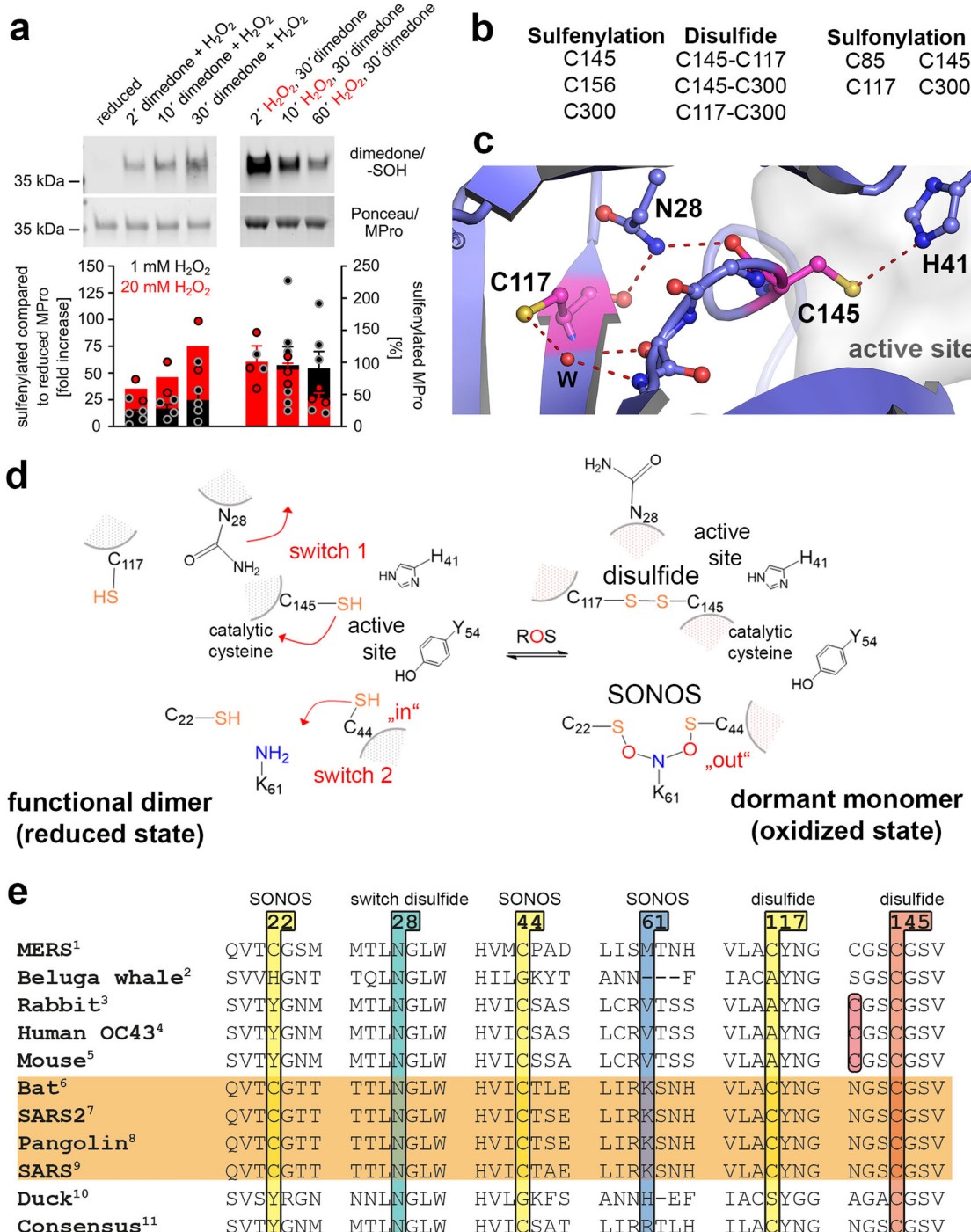

**Fig. 4 | Analysis of redox modifications of SARS-CoV-2 main protease (M^pro).**
**a** Western blot analysis of cysteine sulfenylation (formation of sulfenic acid). Reduced M^Pro was oxidized with either 1 mM (black bars) or 20 mM $H_2O_2$ (red bars). Sulfenylated thiols were trapped by addition of 5 mM dimedone added either simultaneously (left panel) or after pre-incubation with $H_2O_2$ (right panel) for/after indicated time points. Bars represent quantification of sulfenylated thiol/protein using western blots (mean ± s.d.). Data points for 20 mM $H_2O_2$ (in red) from left to right: $n = 2$, $n = 2$, $n = 2$, $n = 3$, $n = 3$, $n = 3$, and for 1 mM $H2O2$ (in black) $n = 5$, $n = 4$, $n = 5$, $n = 2$, $n = 6$, $n = 5$. Source data are provided as a Source Data file (**b**) Mass spec-based redox proteomics analysis of site-specific cysteine modifications in M^pro after oxidation with $H_2O_2$ (disulfides, sulfenylation) as well as concentration dependent sulfonylation (Supplementary Fig. 18). Representative mass spectra are shown in Supplementary Figs. 19, 20. **c** Structure of the M^pro active site and immediate vicinity highlighting the disulfide-forming C145 and proximal C117 interspaced by N28

(pdb code 7KPH). **d** Suggested redox switching mechanism of M^pro. Under oxidizing conditions, catalytic C145 becomes sulfenylated inducing a structural transition (switch 1) that brings C145 and C117 together resulting in formation of the C117-C145 disulfide. This leads to a shift of the oligomeric equilibrium towards the monomeric state. Residues C22, C44 and K61 form the trivalent SONOS bridge (switch 2) that structurally stabilizes the protein under oxidizing conditions. **e** Sequence conservation of disulfide- forming residues C117 and C145 incl. the bridging residue N28 as well as of SONOS bridge- forming residues C22, C44 and K61 in coronaviruses. UniProtKB ID of polyprotein 1ab: [1] K9N7C7, [2] B2BW31, [3] H9AA60, [4] P0C6X6, [5] P0C6X9, [6] E0XIZ2, [7] P0DTD1, [8] A0A6G6A2G5, [9] P0DTD1, [10] A0A0F6WGL5, [11] From 67 sequences. Note that for proteins that do not possess an equivalent cysteine at position of C117, catalytic C145 is found to be in a C-X-X-C145 motif suggesting the possibility of another disulfide switch.

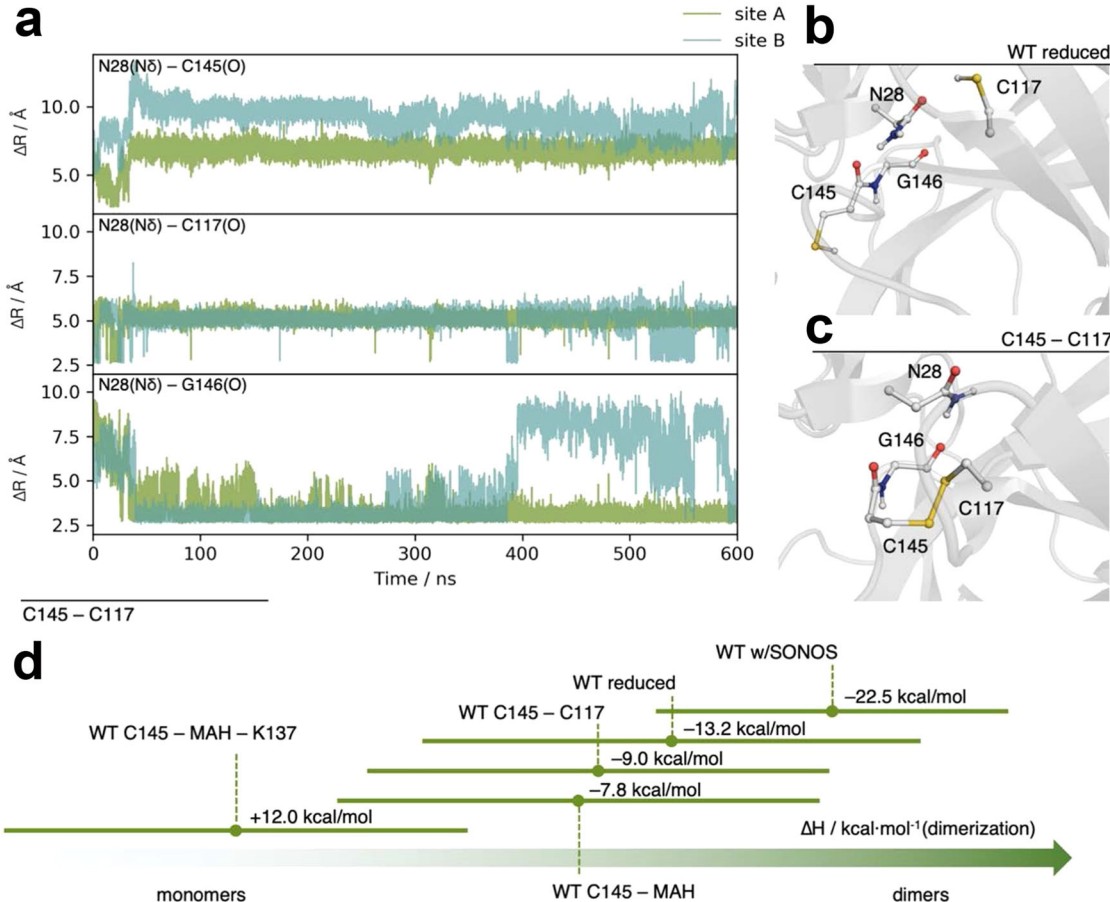

**Fig. 5 | MD simulations of SARS-CoV-2 M^pro in the reduced versus oxidized state (C145- C117 disulfide) and calculated dimer stabilities. a** Selected distances along the 600 ns MD trajectory of M^pro with the disulfide bridge C145-C117 formed on both monomers. After just a few nanoseconds, N28 is displaced from its interaction with the backbone carbonyls of C145 and C117 through the amide moiety. It builds a hydrogen bond to the carbonyl of G146, with occasional interactions to the C117. Snapshots taken from MD trajectories illustrating the interactions of N28. In the reduced protein, N28 has stable interactions with the backbone atoms of C145 and C117 (**b**). Upon disulfide bridge formation (**c**), N28 is flipped and moves to interact with G146. **d** Summary of MMPBSA dimerization enthalpies for the different simulated variants of M^pro. Note the reduced dimerization enthalpies for M^pro containing the C145- C117 disulfide and after covalent binding of MAH to C145 in support of the experimental data. The effect is even clearer when considering the full linkage of the inhibitor (both to C145 and K137), whereby the dimerization process becomes endergonic.

C117 into direct spatial proximity and expels N28 from its original position (switch 1 in Fig. 4d). Our mutagenesis study suggests that the initial oxidation of catalytic C145 to sulfenic acid is central to this structural change (see Fig. 3b). Oxidative insult also leads to formation of the SONOS bridge formed between C22, C44 and K61. In the course of this reaction, C44 flips from the "in" to the "out" conformation (switch 2 in Fig. 4d, Supplementary Fig. 2). Overall, the redox switch mechanism protects M^pro against oxidative damage by forming a C145-C117 disulfide as a stable storage of the catalytic cysteine avoiding an irreversible overoxidation or formation of other kinetically stable modifications. The SONOS bridge structurally stabilizes the monomer that is formed upon oxidation by covalently tethering three structural elements within a monomer.

Sequence analysis indicates that the disulfide-forming residues C145/C117 and adjacent N28 are highly conserved in main proteases from different coronaviruses including SARS-CoV and MERS-CoV suggesting that these orthologs might also subject to redox regulation in vitro (Fig. 4e, Supplementary Figs. 25, 26). To test this hypothesis, we conducted redox-dependent experiments with M^pro from SARS-CoV (Fig. 6). Our analyses clearly reveal that this enzyme, too, is subject to redox regulation in vitro and reversibly switches between the enzymatically active dimer under reducing conditions and the inactive monomer under oxidizing conditions. Based on sequence conservation, the SONOS redox modification seems to be a more specific evolutionary development and is found in M^pro from SARS-CoV and SARS-CoV-2 but not in MERS-CoV or other coronaviruses (Fig. 4e, Supplementary Fig. 25).

### Redox-switch-inspired inhibitors of M^pro
Finally, we tested whether the redox switching can be mimicked by redox-independent crosslinkers as a potential approach to design M^pro inhibitors. As cysteine and lysine residues constitute the genuine redox switches (disulfide and SONOS), we tested homobifunctional (Cys+Cys) and heterobifunctional (Cys+Lys) crosslinkers with warheads targeting thiol and amine functional groups.

From the heterobifunctional compounds tested, maleimidoacetic acid N-hydroxysuccinimide ester (MAH) gave the most interesting results (Fig. 7). Addition of 1 mM MAH resulted in the almost complete dissociation of the M^pro dimer as monitored by both gel filtration and analytical ultracentrifugation experiments (Fig. 7a, b). Enzymatic activity is irreversibly lost after MAH incubation with an $IC_{50}$ value in the micromolar range ($18.2 \pm 1.8 \mu M$) (Supplementary Fig. 27). In contrast, M^pro-targeting drug nirmatrelvir strongly stabilizes the dimer indicating different modes of inhibition for MAH and nirmatrelvir. Mass spectrometric analysis identified catalytic C145 and K137 as the main MAH-binding residues (Fig. 7c, Supplementary

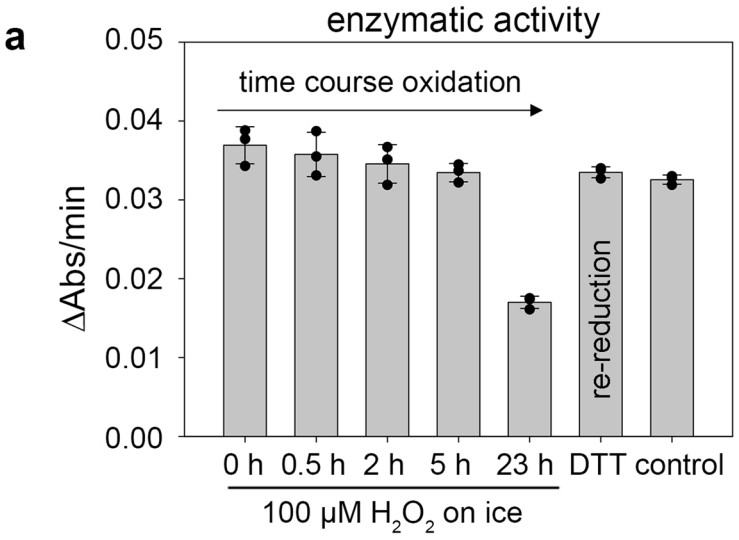

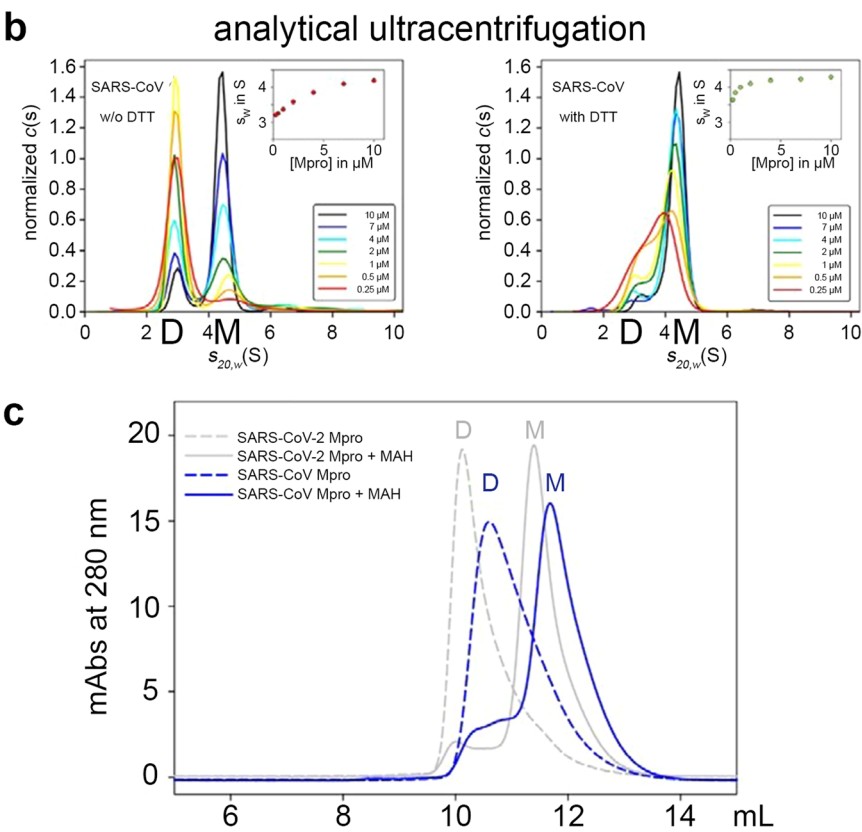

**Fig. 6 | Redox dependence of enzymatic activity and oligomeric equilibrium of SARS-CoV M$^{pro}$ at oxidizing versus reducing conditions. a** Enzymatic activity of M$^{pro}$ after incubation with 100 μM H$_2$O$_2$ on ice for different time points (assay conditions are detailed in the methods section). Note the progressive loss of enzymatic activity over time. Re-reduction of the oxidized enzyme (23 h reaction time) with DTT overnight on ice fully restores enzymatic activity as compared to an enzyme control sample, which was kept on ice for 23 h without adding oxidizing or reducing agents. All measurements were carried out in triplicate and are shown as mean ± s.d. Source data are provided as a Source Data file. Almost identical results were obtained in two independent biological replicates. **b** Sedimentation velocity analysis of SARS-CoV M$^{pro}$ in a concentration range from 0.25 to 10 μM in non-reducing conditions (left panel) versus reducing conditions with DTT (right panel).

The data indicate a redox-dependent monomer ⇔ dimer equilibrium with apparent equilibrium constants of $K_{app}$ ~0.25 μM for the reduced enzyme and of about 2.5 μM for the oxidized enzyme similar to the results obtained for M$^{pro}$ from SARS-CoV-2 (Fig. 1b). Note the slightly decreased stability of the dimer under reducing conditions compared to SARS-CoV-2 M$^{pro}$. Insets show $s_w$ binding isotherms, as calculated from the corresponding c(s) distributions. Abbreviations: M, monomer ($s_{20,w} = 2.9$S); D, dimer ($s_{20,w} = 4.5$ S); O, oligomers (($s_{20,w} = 6.3$ S). **c** Gel filtration analysis of SARS-CoV M$^{pro}$ after addition of 1 mM MAH (blue solid line) and untreated as control (blue dashed line). Abbreviations: D, dimer; M, monomer. Note that addition of MAH results in the almost quantitative formation of the enzymatically inactive monomer similar to M$^{pro}$ from SARS-CoV-2 (grey lines).

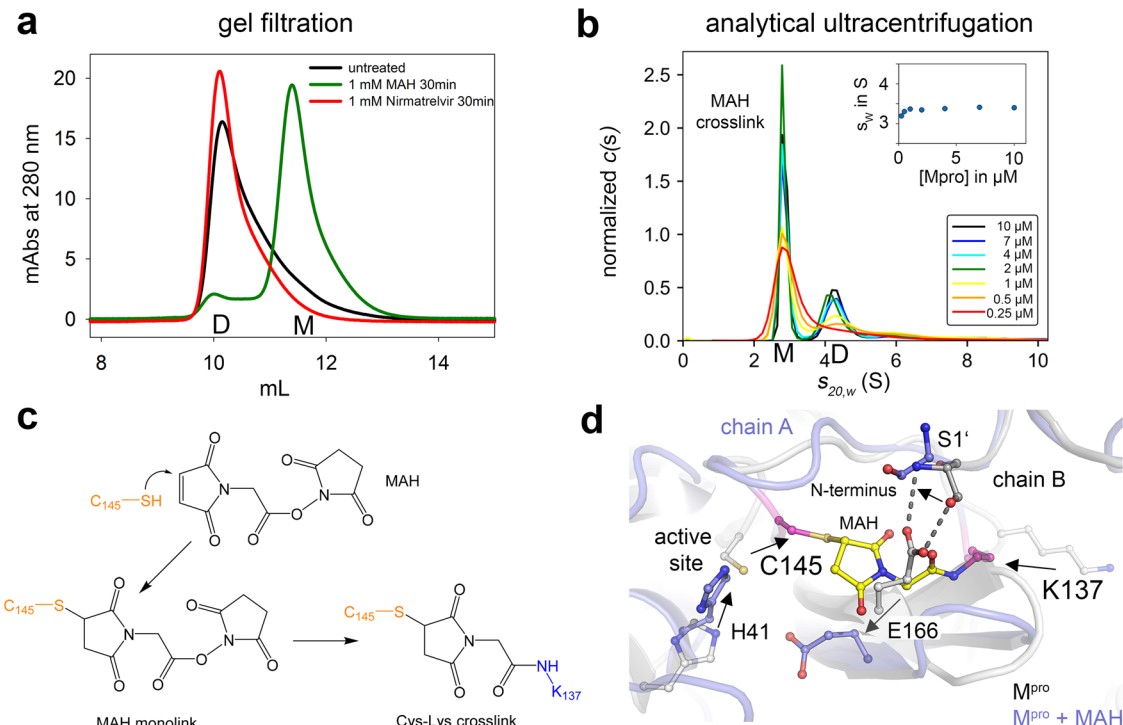

**Fig. 7 | Impact of the heterobifunctional crosslinker MAH on the oligomeric equilibrium of M^pro. a** Gel filtration analysis of M^pro after addition of 1 mM MAH, 1 mM nirmatrelvir (Paxlovid™) and untreated as control. Abbreviations: D, dimer; M, monomer. Note that addition of MAH entails formation of the monomer, whereas addition of nirmatrelvir stabilizes the dimeric state. **b** Sedimentation velocity analysis of SARS-CoV-2 M^pro in a concentration range from 0.25 to 10 µM after incubation with 1 mM MAH. Inset shows the $s_w$ binding isotherm, as calculated from the corresponding c(s) distributions. Abbreviations: M, monomer ($s_{20,w}$ = 2.9 S); D, dimer ($s_{20,w}$ = 4.5 S). Note the preferential destabilization of the dimer in support of the gel filtration experiments (see **a**). **c** Chemical mechanism of MAH crosslinking to M^pro as suggested by mass spec analysis that identified catalytic C145 (thiol) and K137 (amine) as main reaction sites. **d** Structure of the M^pro active site and dimer interface without MAH (grey, pdb code 7KPH) and with MAH covalently bound to C145 and K137 (blue, MD simulation, this study). The MAH moiety is highlighted in yellow, the two crosslinked residues C145 and K137 in magenta. Note that upon covalent binding of MAH the monomer-monomer interaction between E166 and S1' of the neighboring chain, which is critical for dimer formation and stability, is perturbed providing the structural basis for destabilization of the dimeric assembly upon crosslinking. The arrows indicate the displacements of individual residues upon MAH crosslinking.

Fig. 28). As both residues are separated by ~20 Å in all structures determined to date, they could not simultaneously react with MAH without a structural change. Interestingly, the dimer interface, including residues E166 and N-terminal S1' from the second chain, is located right between C145 and K137 rationalizing why a crosslink between the two residues destabilizes the dimer (Fig. 7d). Also, we observed a substantial amount of MAH mono-links to interface residue C300, and, to a lesser extent, to C117. In line with these observations, MAH-induced monomerization is almost absent in variant C145S, whereas reaction of MAH with variants C117S and C300S leads to a complete (C117S) or strong monomerization (C300S) (Supplementary Fig. 29). We investigated the effect of MAH binding to C145 computationally. A model of the crosslinker binding at either C145 alone or bound to both C145 and K137 was built and simulated for a total production time of 3.5 µs (C145-MAH) and 1.0 µs (C145-MAH-K137), respectively. The simulation shows that by covalent bonding of MAH to C145 alone the N28 residue becomes already displaced (Supplementary Fig. 30), breaking the amide interaction with the backbone atoms of C117 and C145, and establishing a new hydrogen bond to the backbone of G146. This in turn facilitates C117 approaching C145 (Supplementary Fig. 30). We further compared the dimerization energies. The MAH/C145 covalent bond structure has a penalty of 5.4 kcal/mol for dimer formation, showing that even before the K137-C145 crosslink formation the oligomerization energetics are already affected (Fig. 5d). Establishment of the second crosslink with K137 increases the binding enthalpy to 12 kcal/mol, which would totally shift the equilibrium to the monomer state in line with the experimental data (Fig. 5d).

A similar dimer-destabilizing effect, albeit not as quantitative as in the case of MAH, is observed upon addition of the homobifunctional crosslinker bismaleimidoethane (BMOE) that preferentially crosslinks C145 and C117 thus mimicking the C145-C117 disulfide-dithiol redox switch (Supplementary Fig. 31, Supplementary Data 4). Interestingly, BMOE is a better inhibitor for M^pro than MAH as the IC50 value is ~10 times smaller than that of MAH and amounts to 1.4 ± 0.2 µM (Supplementary Fig. 27). Using a cell-based SARS-CoV-2 infection model we can demonstrate as proof-of-concept that both MAH as well as BMOE principally exhibit antiviral activity (more than 50-fold reduced virus progeny) with only a mild cytotoxic effect (Supplementary Figs. 32, 33). Quantitative analysis of the dose response curves (virus RNA progeny) shows that the estimated EC50 values are in the millimolar range and thus higher than the IC50 values obtained under in vitro conditions with highly enriched M^pro in buffer (Supplementary Fig. 34). The EC50 value for BMOE is ~10 times smaller (0.8 mM) than that of MAH (7.9 mM) akin to the in vitro studies and estimated IC50 values. The almost identical ratios for EC50 (EC50^BMOE/EC50^MAH) and IC50 values (IC50^BMOE/IC50^MAH) would seemingly suggest that antiviral activity of these compounds is reflecting inhibition of M^pro. Admittedly, we cannot rule out that these compounds inhibit viral replication by "off-target" effects on other viral proteins, e.g. the papain-like protease, or host proteins. Our systematic analysis of MAH and BMOE cytotoxicity implies that the antiviral activity of both compounds mostly reflects inhibition of M^pro but cytotoxicity is also contributing to a smaller extent (Supplementary Fig. 35). Both compounds exhibit a mild cytotoxic effect in the concentration regime relevant for the dose response analysis, in particular BMOE, but become clearly cytotoxic at higher

concentrations. We would like to note that cytotoxicity of both compounds might be more pronounced in other cell lines.

The finding of $EC_{50}$ values for BMOE and MAH in the mM range is not surprising given the high reactivity and lack of selectivity of the thiol and amine warheads of MAH (maleimide, NHS ester) and BMOE (maleimide). Nitril or ketoamide warheads as used for e.g. nirmatrelvir or other promising $M^{pro}$-targeting inhibitors are likely to be more selective, less toxic and chemically less reactive warheads binding to cysteine residues[12,23]. While the amine function of lysine residues is typically considered a non-optimal target group for covalent drugs, very recent studies highlighted boronates (o-aminomethyl phenylboronic acid) or salicylaldehydes as promising compounds that reversibly bind to lysines in covalent fashion[41–43]. We envisage the design of a covalent $M^{pro}$ inhibitor with multiple covalently binding warheads as a promising direction to increase selectivity and efficacy using the redox switch mechanism described here as a blueprint. These could target the catalytic cysteine C145 and C117, which form the disulfide-dithiol switch, and/or K137 as these residues are conserved in many Coronavirus main proteases (Supplementary Fig. 36). As proof-of-concept, we found that MAH inhibits $M^{pro}$ from SARS-CoV in vitro in the same way as $M^{pro}$ from SARS-CoV-2 leading to a destabilization of the dimer (Fig. 5c). Alternatively, SONOS residue C44 or glutathionylation site C300[44] at the dimer interface, both proximal to catalytic C145, could be target sites for covalently binding warheads.

In summary, we have reported on a hitherto unknown mode of redox regulation of SARS-CoV-2 $M^{pro}$ in vitro that protects the redox-vulnerable catalytic cysteine and the structural integrity of the protein under oxidative stress conditions that are known to accompany SARS-CoV-2 infection[45]. The in vivo relevance of the redox switches remains to be confirmed (or ruled out) in future studies, yet it seems that the detected time scales of the underlying processes and oxidation conditions are in principle compatible with physiological oxidative stress conditions. The unusually high abundance of cysteine residues (4%) distributed over the protein molecule would ensure resistance to oxidative stress conditions not only by scavenging of ROS but by multiple sophisticated redox switches that protect against over-oxidation of the catalytic cysteine (disulfide-dithiol with C117) and against destabilization of the three-dimensional protein structure by establishing a trivalent SONOS bridge that tethers three spatially proximal structural units via residues C22, C44 and K61.

Additional redox regulatory mechanisms might involve glutathionylation of cysteines (C300) as recently reported[44]. Glutathionylation and disulfide formation might function synergistically as both drive dissociation of the functional dimer into the nonfunctional monomer. Also, a glutathionylation of C300 as reported might prevent formation of a kinetically stable C145-C300 disulfide crosslink. The detected redox switches in the main protease seem to be widespread amongst coronaviruses and it is likely that other viral cysteine proteases such as the papain-like protease have evolved similar defense mechanisms. As the redox switching can be mimicked by non-redox chemistry, this offers opportunities in the design of inhibitors targeting viral cysteine proteases using the redox switches as common druggable sites.

## Methods
### General information
The protein concentration was determined by UV/Vis spectroscopy using the absorbance signal at 280 nm and the molar extinction coefficient ($\varepsilon_{MPro} = 32{,}890\ M^{-1}cm^{-1}$), which was calculated according to Gill and von Hippel[46].

### Mutagenesis
Variants of SARS-CoV-2 $M^{pro}$ were generated by site-directed mutagenesis PCR using the KLD enzyme mix (New England Biolabs) and expression vector pGEX-6P1 NSP5 (https://mrcppureagents.dundee. ac.uk/reagents-view-cdna-clones/703227). Correctness of the introduced mutations was confirmed by complete sequencing of the gene.

The primers are listed in Supplementary Table 6.

### Expression
For recombinant expression, vector pGEX-6P1 NSP5 containing the $M^{pro}$ gene was transformed into BLR(DE3) chemically competent *E. coli* cells (Novagen, Merck Biosciences, affiliate of Merck KGaA, Darmstadt, Germany), containing an pREP4 plasmid, according to Inoue et al.[47]. The bacteria were grown in LB media[48] containing 50 µg/mL kanamycin sulfate and 100 µg/µL carbenicillin (disodium salt) at 37 °C until an optical density at 600 nm ($OD_{600}$) of 0.6 was reached. The cells were then incubated at 18 °C for 30 min until an $OD_{600}$ of 0.8. Subsequently, gene expression was induced by addition of 500 µM isopropyl-ß-D-thio-galactopyranoside (IPTG) for ~20 h at 18 °C. The cells were harvested by centrifugation at 5750 x *g* and either directly used or flash frozen in liquid nitrogen and stored at −80 °C until usage. $M^{pro}$ from SARS-CoV was expressed using the original expression plasmid from the Hilgenfeld lab[12].

### Protein purification
All purification steps were performed at 4 °C or on ice. Cells were resuspended in buffer A (20 mM Tris/HCl pH 7.8, 5 mM imidazole, 150 mM NaCl, 1 mM dithiothreitol), supplemented with 100 µM phenylmethanesulfonyl fluoride, 0.5 mg/mL lysozyme (AppliChem GmbH Darmstadt, Germany), 5 mM $MgCl_2$ and 5 µg/mL DNaseI (Thermo Fisher Scientific Braunschweig, Germany), and subsequently lysed by five passages through a LM10 Microfluidizer® High Shear Fluid Homogenizer (Microfluidic Corp, Newton, MA, USA).

Next, the lysate was centrifuged at 75,000 x *g* for 30 min and the thereby obtained supernatant was loaded onto a HisTrap™ HP 3x5 mL column (GE Healthcare Munich, Germany). The His6x-$M^{pro}$ fusion protein was then eluted with buffer B (20 mM Tris/HCl pH 7.8, 300 mM imidazole, 150 mM NaCl, 1 mM dithiothreitol) and subsequently dialyzed against 2 L buffer A overnight. To remove the His-tag, 1 mg PreScission Protease was added per 10 mg $M^{pro}$. The cleaved His-tags and non-cleaved protein were then separated from untagged $M^{pro}$ via affinity chromatography as described above. The PreScission Protease was removed via an additional affinity chromatography step, employing a GSTrap™ HP column (GE Healthcare Munich, Germany). $M^{pro}$ was then treated with 1 mM EDTA and subjected to size exclusion chromatography using a HiLoad 16/60 Superdex 75 prep grade gel filtration column (GE Healthcare, Munich) in buffer C (20 mM Tris/HCl pH 7.8, 150 mM NaCl, 1 mM dithiothreitol). The purified protein was either directly used for experiments or supplemented with 20% (*v/v*) glycerol, flash-frozen with liquid nitrogen and stored at -80°C until usage.

### Steady-State kinetics
For steady-state kinetic analysis of enzymatic activity of $M^{pro}$ wild-type and variants under reducing and oxidizing conditions, the cleavage of an artificial $M^{pro}$ peptidic substrate (Ac-Abu-Tle-Leu-Gln-AMC, Biosynth Carbosynth, Switzerland) was monitored spectrophotometrically at 380 nm in a UV−Vis spectrometer (V-750, Jasco GmbH, Germany).

Prior to the kinetic measurements, $M^{pro}$ in assay buffer (20 mM Tris pH 7.3, 100 mM NaCl, 1 mM EDTA) was incubated with either 1 mM $H_2O_2$ or 1 mM dithiothreitol (DTT) for 2 h. For measurements of reactivation, the protein was first incubated with 1 mM $H_2O_2$ for 2 h on ice, whereupon the $H_2O_2$ was removed using a 5 mL HiTrap Desalting column (GE Healthcare Munich, Germany). Subsequently, oxidized $M^{pro}$ was (re)-reduced by incubation with 20 mM DTT for 3 or 20 h. The reaction was started by adding 1 µM $M^{pro}$ to a preincubated reaction mix (200 µL) containing 200 µM peptide substrate and either 1 mM $H_2O_2$ or 1 mM DTT in assay buffer at 20 °C. The change in absorption was continuously monitored at 380 nm ($\varepsilon_{AMC} = 2400\ M^{-1}cm^{-1}$). Initial

rates were estimated by linear regression of the absorbance signal over the first 10 s of the measurements or, in cases in where substrate activation was observed, using Eq. 1

$$A_{340}(t) = A_0 - \triangle ss \cdot t + \frac{\triangle ss - \triangle 0}{k_{obs}} \cdot \left[ 1 - \exp(-k_{obs} \cdot t) \right] \quad (1)$$

in which $A_0$ denotes the starting absorbance at 380 nm, $\Delta ss$ the absorbance changes at steady-state (steady-state rate), $\Delta 0$ the absorbance changes at $t = 0$ (initial rate), and $k_{obs}$ the first-order rate constant of activation.

For the estimation of the IC50 values for bifunctional crosslinkers MAH and BMOE, enzymatic activity was measured under identical conditions as indicated above but after pre-incubation of $M^{pro}$ with varying concentrations of crosslinkers for 30 min on ice. Kinetic data were analysed with Eq. 2:

$$v(A) = V_{max} \cdot \frac{IC50^n + [A]^n}{[A]^n} \quad (2)$$

in which $V_{max}$ is the maximal activity in the absence of crosslinkers, $[A]$ is the applied crosslinker concentration, IC50 is the concentration of crosslinker with 50% inhibition of enzymatic activity and $n$ is the Hill coefficient.

### Secondary structure and thermal unfolding analysis

To analyze secondary structure contents and thermal stability of $M^{Pro}$ wild-type and variants, far-UV circular dichroism (CD) spectra and thermal unfolding data were collected using a circular dichroism spectrometer (Chirascan, Applied Photophysics, UK). Far-UV CD spectra were collected in a range of 195–260 nm and using a concentration of 0.2 mg/ml protein treated with either 1 mM $H_2O_2$ or 1 mM DTT for 2 h on ice, in 100 mM $Na_2HPO_4$ pH 7.8, with a step size of 1 nm and at least 20 accumulations for 0.5 s per wavelength. Secondary structure contents were calculated using the CDNN software[49].

Thermal unfolding was monitored at a wavelength of 222 nm in a temperature range from 20–95 °C (real sample temperature was determined using a temperature probe) with a ramping speed of one °C/min. Each temperature data point was collected for 10 s.

### Analytical ultracentrifugation (AUC)

Sedimentation velocity experiments (SV) were performed in analytical ultracentrifuges ProteomeLab XL-I or Optima AUC (Beckman Coulter, USA) at 50,000 rpm and 20 °C using An-50 Ti rotors. Concentration profiles were measured using the absorption scanning optics at 230 nm with 3 or 12 mm standard double sector centerpieces filled with 100 μl or 400 μl sample, respectively. Stock solutions of SARS-CoV-2 $M^{pro}$ and mutants thereof were dialyzed overnight against buffers containing 0.1 M NaCl and 20 mM Tris pH 7.3 in the absence (AUC buffer) or presence of 1 mM DTT (AUC buffer + DTT). After dilution to concentrations in the range of 0.25 to 10 μM, samples were allowed to equilibrate for 23 h at room temperature before SV analysis, since it has been found that the monomer-dimer equilibrium of $M^{pro}$ is slow on the time-scale of centrifugation[12].

To test whether $M^{pro}$ can be regenerated after removal of DTT, $M^{pro}$ stock solution was first dialyzed overnight against AUC buffer, afterwards diluted to 0.25 to 10 μM in AUC buffer + DTT. Samples were allowed to equilibrate for 23 h at room temperature before SV analysis. In a second experiment following dialysis against AUC buffer and dilution in AUC buffer to final concentration of 0.5 to 10 μM and another 23 h of incubation at room temperature, a 20 mM DTT stock solution (in AUC buffer) was added to a final concentration of 1 mM. $M^{pro}$ was further incubated for 23 h at room temperature before SV analysis. The addition of DTT stock solution resulted in a slight dilution of the samples (0.47 to 9.5 μM).

For $M^{pro}$ oxidation by 1 mM $H_2O_2$, DTT-containing stock solutions were transferred to AUC buffer by ZEBA Spin Desalting Columns (Thermo Scientific), diluted to 25 μM and incubated with 1 mM $H_2O_2$ for 2 hours on ice. Subsequently, $H_2O_2$ was removed by ZEBA Spin Desalting Columns, samples were centrifuged for 20 min at 15,000 x $g$ and protein concentrations were determined spectrophotometrically. Proteins were diluted to 0.25 to 10 μM and were analysed by SV about 2 h after dilution and at least 3 hours after $H_2O_2$ treatment. A similar protocol was applied for chemical crosslinking of $M^{pro}$ with heterobifunctional crosslinker maleimidoacetic acid N-hydroxysuccinimide ester (MAH), except that $H_2O_2$ treatment was replaced by incubation with MAH for 30 min on ice and 50 mM sodium dihydrogen phosphate pH 7.3 was used as a buffer. Untreated proteins were analysed in 50 mM sodium dihydrogen phosphate pH 7.3 as a control.

For $M^{pro}$ oxidation by 100 μM $H_2O_2$, DTT-containing stock solutions were dialyzed for 4–5 h against AUC buffer, diluted to 25 μM $M^{pro}$ and incubated with 100 μM $H_2O_2$ for 16 h on ice. Subsequently, $H_2O_2$ was removed by a ZEBA Spin desalting column. One half of the sample was dialysed for 4 h against AUC buffer and the other half against AUC buffer + DTT. After dilution to concentrations in the range of 0.25 to 10 μM with AUC buffer or AUC buffer + DTT, respectively, samples were allowed to equilibrate for 23 h at room temperature before SV analysis.

For data analysis, a model for diffusion-deconvoluted differential sedimentation coefficient distributions (continuous c(s) distributions) implemented in the program SEDFIT[50] was used. Partial specific volume and extinction coefficient of the protein as well as buffer density and viscosity, were calculated from amino acid and buffer composition, respectively, by the program SEDNTERP[51] and were used to calculate protein concentration and correct experimental s-values to $s_{20,w}$.

Signal-averaged s-values $s_w$ were obtained by integration of the c(s) distributions in the s-value range where monomers and dimers were observed using the program GUSSI[52], and plotted as a function of concentration to obtain binding isotherms for the monomer-dimer equilibrium of SARS-CoV-2 $M^{pro}$.

### Analytical size exclusion chromatography

To analyze the distribution of higher oligomers/aggregates, dimers and monomers of $M^{Pro}$ under reducing and oxidizing conditions, 25 μM $M^{Pro}$ in assay buffer (20 mM Tris pH 7.3, 100 mM NaCl, 1 mM EDTA) were pre-incubated with either 1 mM $H_2O_2$ or 1 mM DTT for 1–5 h on ice and then loaded onto a Superdex 75 increase 10/300 GL column (GE Healthcare, Munich) via ÄKTA Pure 25 M (GE Healthcare, Munich) at 6 °C. The peak heights and integrals were analysed using the UNICORN 7.1 software.

### Crystallization and Cryoprotection

$M^{pro}$ crystals were grown at 20 °C using the hanging-drop vapor diffusion method with a reservoir solution containing 0.1 M MES pH 6.5, 8-10% PEG 3350, 1.5% DMSO and 0.1 M sodium citrate. 1 μL of reservoir solution was mixed with 1 μL protein solution containing 10 mg/mL $M^{pro}$ in buffer C (20 mM Tris/HCl pH 7.8, 150 mM NaCl, 1 mM DTT). Cryoprotection was carried out using 20% ($v/v$) glycerol in well solution, soaking the crystals for up to 90 seconds.

### X-ray data collection, processing and model building

Diffraction data of $M^{Pro}$ single crystals (variants C44S, Y54F, K61A) were collected using synchrotron radiation at beamline P14 of the DESY/EMBL Hamburg, Germany at a wavelength of 0.827 or 0.976 Å. Data were collected at cryogenic temperature (100 K) with an EIGER 16 M detector. Processing using anisotropic cut-off limits was performed using autoPROC[53], which calls on the XDS package[54], the CCP4 suite of programs[55] and STARANISO[56].

Subsequent refinement and model building was performed employing Phenix.REFINE[57] and COOT[58]. Phasing was performed using MOLREP[59] using the published M$^{Pro}$ structure (PDB ID 6LU7) as starting model. The geometry of the structural models was validated using MOLPROBITY[60]. Structural representations were prepared using PyMOL[61]. Crystallographic statistics are provided in Supplementary Table 1.

The refined structural protein models and corresponding structure-factor amplitudes have been deposited under PDB accession codes 7ZB6 (C44S), 7ZB7 (Y54F) and 7ZB8 (K61A). The Ramachandran statistics are 91.45% in the favoured, 8.55% in the allowed and 0% in the outlier region for 7ZB6; 98.03%, 1.64% and 0.33% for 7ZB7; and 87.99%, 9.87% and 2.14% for 7ZB8.

## Redox proteomics

M$^{Pro}$ was analyzed to study **a**) sulfenylation after Western transfer (using the BioRad system) and **b**) to determine site-specific oxidative modifications via mass spectrometry. For a), M$^{Pro}$ was incubated for 30 min with 10 mM DTT on ice. DTT was removed using Zeba spin columns (Thermo Scientific). An amount of 0.5, 1.0, or 5.0 µg of reduced MPro were incubated in 20 mM Tris/HCl pH 7.8, 150 mM NaCl on ice for different periods of time (2–60 min) with 1 or 20 mM $H_2O_2$. Dimedone (5 mM) was either added simultaneously or after pre-incubation with $H_2O_2$. The remaining thiols were blocked by the addition of 100 mM NEM (N-Ethylmaleimide). After SDS-PAGE, proteins were transferred to nitrocellulose membranes and sulfenylation was visualized by anti-dimedone antibodies (Pineda antibodies, Berlin, Germany)[62]. b) Similar treated M$^{Pro}$ (see above) was directly applied for mass spectrometry or Coomassie-stained proteins were prepared for mass spectrometry. In addition, M$^{Pro}$ was treated with DTT (1 mM, 2 h on ice) and $H_2O_2$ (100 µM, 1 mM, 2 h on ice and 20 mM, 30 minutes on ice). The thereby obtained protein was either used for crosslinking after gel-filtration and buffer exchange to phosphate buffered saline including 1 mM EDTA, or cysteines were blocked with 100 mM NEM. Crosslinking of M$^{Pro}$ was carried out by incubating 10 µg M$^{Pro}$ in a total volume of 20 µl phosphate buffered saline including 1 mM EDTA and 1 mM crosslinker (BMOE or MAH) for 30 minutes at 22 °C. The reaction was stopped by adding 1 µl 1 M Tris pH 7.5 and 4x sample buffer including DTT followed by polyacrylamide gel separation. M$^{Pro}$ was separated under reducing (+150 mM DTT) or non-reducing conditions (without DTT) in polyacrylamide gels and stained with Coomassie brilliant blue essentially as described[63]. Protein-containing bands were cut out of the gel and - depending on the experiment - reduced with DTT and alkylated with iodoacetamide as described[63] or only alkylated with iodoacetamide. Finally, the M$^{Pro}$ samples were in-gel digested with 0.1 µg chymotrypsin in 23 µl of 100 mM Tris-HCl and 10 mM CaCl$_2$ in water (pH 7.8) overnight. Resulting peptides were extracted from the gel and resuspended in 0.1% trifluoroacetic acid as previously described[63]. Subsequently, peptides were separated using an Ultimate 3000 rapid separation liquid chromatography system (Thermo Fisher Scientific) as described before[62]. Briefly, peptides were loaded on a 2 cm length trap column for 10 minutes and subsequently separated over 54 minutes on a 20 cm C18 analytical column. Eluting peptides were directly sprayed into the mass spectrometer via a nanosource electrospray interface. An Orbitrap Fusion Lumos (Thermo Fisher Scientific) mass spectrometer, operated in positive mode, was used for the analysis of M$^{Pro}$ peptides. First, precursor spectra were recorded in the orbitrap in profile mode (resolution 60,000, scan range 400–1800 m/z, maximum injection time 50 ms, AGC target 100,000). Thereafter, 2–10-fold charged precursors were selected by the quadrupole (isolation window 1.6 m/z, minimum intensity 50,000) fragmented via higher-energy collisional dissociation and analyzed in the orbitrap and afterwards newly selected and fragmented with collision-induced dissociation and analysis in the orbitrap. Fragment spectra were recorded in centroid mode (resolution 30000, maximum injection time 120 ms, AGC target 50,000, scan range: auto). The cycle time

was 2 seconds, already fragmented precursors were excluded from isolation for the next 60 seconds. Peptide and crosslink identification were carried out with MaxQuant version 2.0.3.0 (Max-Planck Institute for Biochemistry, Planegg, Germany) with standard parameters if not stated otherwise. The M$^{Pro}$ amino acid sequence was used as search template, following variable modifications were considered: acetylation (N-terminus), oxidation (methionine), carbamidomethylation (cysteine), glutathionylation (cysteine), di-oxidation (cysteine), tri-oxidation (cysteine). Depending on the analysed samples, additional modifications with NEM (cysteine), NEM + water (cysteine) and dimedone (cysteine) were considered. Crosslink searches were enabled by screening for disulfides (−2.0157), links between cysteines by BMOE (+220.0484) and BMOE + water (+238.059), links between cysteines with MAH (+137.0113) and MAH + water (+155.0219) and between cysteines and lysines with MAH (+137.0113) and MAH + water (+155.0219). The match-between-runs option was enabled for searches for BMOE and MAH crosslinks, proteins and peptides were identified at a false discovery rate of 1%. Data analysis was carried out in excel based on "evidence" and "crosslinkMsms" tables. Here, identified spectra were counted for each run or intensities for all modified peptide variants were summed up per analysed sample. Disulfide crosslinks were accepted upon the following criteria: found after treatment with 100 µM $H_2O_2$ in at least two independent experiments, a minimum of 10 identified spectra in sum in 11 different samples (2 h 100 µM $H_2O_2$ $n = 2$, 2 h 1 mM $H_2O_2$ $n = 3$, 20 min 20 mM DTT $n = 3$, 2 h 1 mM DTT $n = 3$).

## Cell culture

Vero E6 cells (Vero C1008) were maintained in Dulbecco's modified Eagle's medium (DMEM with GlutaMAX$^{TM}$, Gibco) supplemented with 10% fetal bovine serum (Merck), 50 µg/mL streptomycin (Gibco), 50 units/mL penicillin, 10 µg/mL ciprofloxacin (Bayer) and 2 µg/mL tetracycline (Sigma) at 37 °C in a humidified atmosphere with 5% $CO_2$.

## MAH/BMOE treatment and SARS-CoV-2 infection

20,000 cells per well were seeded into 24-well-plates and incubated overnight at 37 °C. Cells were treated with varying concentrations of either MAH (Sigma-Aldrich, diluted in PBS, pH adjusted to 7.5) or BMOE (Sigma-Aldrich, diluted in PBS + 5% DMSO, pH adjusted to 7.5) or the PBS control for 1 h before infection, and then throughout the time of infection, using medium containing 2% fetal bovine serum (FBS). Cells were infected with virus stocks corresponding to 1*10$^7$ RNA-copies of SARS-CoV-2 (=30 focus forming units, FFU) and incubated for 48 h at 37 °C, as described[64]. Cell morphology was assessed by bright field microscopy.

## Quantification of lactate dehydrogenase release to determine cytotoxicity

The release of lactate dehydrogenase (LDH) into the cell culture medium of MAH- or BMOE-treated cells was quantified by bioluminescence using the LDH-Glo$^{TM}$ Cytotoxicity Assay kit (Promega). 10% ($v/v$) Triton X-100 was added to untreated cells for 15 min to determine the maximum LDH release, whereas the medium background (= no-cell control) served as a negative control. Percent cytotoxicity was calculated using the following formula and reflects the proportion of LDH released to the media compared to the overall amount of LDH in the cells.

$$Cytotoxicity(\%) = 100 \times \frac{(Experimental\ LDH\ Release - Medium\ Background)}{(Maximum\ LDH\ Release\ Control - Medium\ Background)} \quad (3)$$

## Quantitative RT-PCR for SARS-CoV-2 quantification

For RNA isolation, the SARS-CoV-2-containing cell culture supernatant was mixed with the Lysis Binding Buffer from the MagNA Pure LC Total

Nucleic Acid Isolation Kit (Roche) to inactivate the virus. The viral RNA was isolated as described[64] and quantitative RT-PCR was performed according to a previously established RT-PCR assay[65], to quantify SARS-CoV-2 RNA yield. The amount of SARS-CoV-2 RNA determined upon infection without any treatment was defined as 100%, and the other RNA quantities were normalized accordingly. A two-sided unpaired Student's t-test was calculated using GraphPad Prism 9.

Dose-response experiments were analysed using Eq. 4

$$E(A) = \frac{E\,max}{1 + \left(\frac{EC50}{[A]}\right)^{n_H}} \qquad (4)$$

in which E is the response to treatment with crosslinker MAH or BMOE (A), Emax is the maximal response (100%), EC50 is the concentration of inhibitor with 50% response, [A] is the inhibitor concentration and $n_H$ the Hill coefficient.

## Immunofluorescence analyses
Vero E6 cells were treated/infected as indicated. After 48 hours of SARS-CoV-2 infection, the cells were washed once in PBS and fixed with 4% formaldehyde in PBS for 1 hour at room temperature. After permeabilization with 0.5% (v/v) Triton X-100/PBS for 30 min and blocking in 10% FBS/PBS for 10 min, primary antibodies were used to stain the SARS-CoV-2 Spike (S; GeneTex #GTX 632604, 1:2000) and Nucleoprotein (N; Sino Biological #40143-R019, 1:8000) overnight. The secondary Alexa Fluor 488 donkey anti-mouse IgG (Thermo Fisher Scientific, Cat# A21202) and Alexa Fluor 546 donkey anti-rabbit IgG (Thermo Fisher Scientific, Cat# A10040) antibodies were added together with DAPI for 1 h at room temperature (1:500, diluted in 10% FBS/PBS). Slides with cells were mounted with DAKO and fluorescence signals were detected by microscopy (Zeiss Axio Scope.A1).

## Immunoblot analysis
Vero E6 cells were treated/infected as indicated. After 48 hours of SARS-CoV-2 infection, the cells were washed once in PBS and then harvested in RIPA lysis buffer (20 mM TRIS-HCl pH 7.5, 150 mM NaCl, 10 mM EDTA, 1% (v/v) Triton-X 100, 1% deoxycholate salt (w/v), 0.1% (v/v) SDS, 2 M urea), supplemented with protease inhibitors. After sonication and equalizing the amounts of protein, samples were separated by SDS-PAGE. To determine the presence of viral proteins, the separated proteins were transferred to a nitrocellulose membrane, blocked in 5% (w/v) non-fat milk in TBS-T for 1 h, and incubated with primary antibodies at 4 °C overnight, followed by incubation with peroxidase-conjugated secondary antibodies (donkey anti-rabbit or donkey anti-mouse IgG, Jackson Immunoresearch). The SARS-CoV-2 Spike (S; GeneTex #GTX 632604, 1:1000) and Nucleoprotein (N; Sino Biological #40143-R019, 1:5000), and GAPDH (abcam ab8245, 1:5000) were detected using Immobilon Western Substrate (Millipore).

## Analysis of M^Pro sequence conservation
The dataset used for analysis was generated using the replicase polyprotein 1ab of SARS-CoV2 (Uniprot-ID P0DTD1) as query for *blastp*. A total of 67 1ab polyprotein sequences were selected from the resulting search for further analysis. These were truncated to the M^Pro-sequence. Alignment and tree-generation were performed using ClustalOmega[66]. Tree visualization was performed using iTOL[67]. Alignment analysis was performed using Jalview[68].

## Quantum chemical calculations - Parametrization of NOS and SONOS
Given that there are no available parameters in the standard Amber forcefield sets for any cysteine-lysine covalent linkage, we carried out a parameterisation of the NOS and SONOS bonds. In a first stage, we

built the parameters for the single NOS bond using a small model system ($CH_3NOSCH_3$). We minimized the structure, employing the Gaussian 16 RevA.03 software package[69] at the B3LYP-D3(BJ) level of theory[70–72] and the def2-SVP basis set[73,74]. The partial atomic charges were assigned using the RESP procedure[75] at the HF/6-31 G* level. The Seminario approach (implemented in the CartHess2FC tool provided with the Amber20 program package) was then employed for the R enantiomeric form of the NOS, in order to obtain the parameters for the NT-OS-S angle[76]. We then performed the scans along the dihedral angles CT-NT-OS-S, H1-CT-NT-OS, H1-CT-S-OS, NT-OS-S-CT and OS-NT-OS-S, at the same DFT level previously mentioned. At this point, a genetic algorithm was employed in order to fit the dihedral angles potentials (http://www.ub.edu/cbdd/?q=content/small-molecule-dihedrals-parametrization) to the DFT values, considering the non-bonded interactions. All the other parameters were provided by the antechamber tool for Amber type atoms.

For the SONOS, we made use of the crystal structure coordinates of the C22, C44 and K61 residues present in the X-ray structure (PDB: 7JR4). We capped the backbones at the C and N atoms saturating them with hydrogens. We then optimized the system at the aforementioned DFT level, constraining all the non-hydrogen backbone atoms to their crystallographic positions. At this stage, we obtained the partial atomic charges at HF/6-31G* level, subtracting the values of the added cap H atoms. The antechamber tool was used to generate the Amber parameters to the oxidized lysine residue and the disulfide bridge atom types were used for the cysteines bound to the lysine.

## Starting structure for the disordered loop of the SONOS containing M^pro dimer
The missing residues (#46, #47 and #48) were modelled by setting up a system formed by the mentioned residues and the neighbouring #45 and #49, which were capped at the terminal C and N, saturating them with hydrogen atoms. The system was then minimized using the semiempirical PM6 Hamiltonian[77] constraining the non-hydrogen atoms of residues #45 and #49. The obtained cartesian coordinates of the minimized non-hydrogen atoms were then manually added to the X-ray crystal structure (PDB: 7JR4).

## Modelling of the MAH containing amino acid
In order to model the MAH-C145 and K137-MAH-C145 bonded systems, a model system was built capping at the K137 and C145 backbone C and N atoms, saturated with hydrogens. The H atoms were relaxed at the B3LYP-D3(BJ)/ def2-SVP level of theory. The partial atomic charges were assigned using the RESP procedure, subtracting the values of the added cap H atoms, at HF/6-31G* level. The forcefield parameters were assigned with the antechamber tool using Amber atom types.

The starting structure for the MAH containing inhibitor is the reduced structure. Since the C145 and K137 residues are pointing away from each other, a constrained minimization was first performed in order to create an adequate structure. First, all the H atoms and the residues 1, 2 and 145 were minimized for 2000 cycles, 1000 cycles with steepest descent and 1000 cycles with conjugate gradient, restraining the rest of the atoms with a restraint of 1000.0 kcal/mol/Å². A second minimization was then performed for 10,000 cycles (5000 cycles with steepest descent and 5000 cycles with conjugate gradient), restraining the backbone atoms with a force constant of 10 kcal/mol/Å². We have herein restrained the S and N atoms of CYS and LYS residues to a distance of 6.5 Å employing a potential with a force constant of 350.0 kcal/mol/Å² so that the two atoms that covalently bind to the MAH inhibitor point to each other. This leads to the starting structure employed for the simulation of the covalently attached MAH. In the case of the singly linked C145-MAH starting structure, the molecule was inserted manually, followed by the standard preparation protocol used for the other MD runs.

## Replica-exchange constant pH simulations

Molecular dynamic simulations were performed setting HIS41, GLU47, ASP48, LYS61, HIS64, HIS163 and HIS164 as titratable. Every other GLU, ASP and LYS are set as charged groups and the protonation states for the other histidines were set as: HID80, HIE172 and HIE246. We have performed the simulations for the reduced, disulfide and SONOS-containing dimeric and monomeric systems.

The RE-cpH simulations were performed with the AMBER20 software package[78,79], using sander and pmemd, employing the ff10 force field[80,81]. The protein was set in a cuboid periodic box leaving an 8 Å distance between the protein atoms and the periodic box wall. TIP3P water molecules neutralized with Na+ and Cl- counter ions were included[82]. The cut off for non-bonded interactions was set to 8 Å, employing particle-mesh Ewald summation with a fourth-order B-spline interpolation and a tolerance of $10^{-5}$. The non-bonded list was updated every 50 fs, and the MD time step was set to 2 fs, employing the SHAKE algorithm to constrain bonds involving hydrogen atoms[83].

The H atoms and residues #44, #46, #47 and #48 of the system were first minimized for 2000 cycles (1000 with steepest descendent and 1000 with conjugate gradient) by restraining the rest of the atoms with a 1000 kcal/mol/Å² force constant. The system was then minimized for another 3000 cycles (1000 with steepest descendent and 2000 with conjugate gradient) restraining the non-hydrogen backbone atoms of the protein with a 10 kcal/mol/Å² force constant. Finally, the system was minimized for 10,000 cycles (2000 with steepest descendent and 8000 with conjugate gradient), allowing all the atoms to relax.

The system was heated from 0 to 300 K in the first 800 ps of an overall 1 ns run, using a NVT ensemble, employing Langevin dynamics with a collision frequency of 5 ps⁻¹. The system was then equilibrated for 1 ns in the NPT ensemble at 300 K and with isotropic position scaling and a relaxation time of 5 ps. The production phase is done using 16 replicas employing the same ensemble and parameters as in the equilibration phase. The production is carried for 128 ns, attempting to change the protonation state every 200 fs and attempting replica exchanges every 4 ps. The heating and production phases were performed using graphics processing unit (GPUs)[84,85].

## Molecular dynamic simulations

The molecular dynamic simulations were performed setting all the GLU, ASP and LYS residues charged and the histidines in the following protonation states: HID41, HID64, HID80, HIE163, HIE172, HIE246 and HIP164.

The simulations were performed with the AMBER20 software package, using sander and pmemd, employing the ff99SB force field[86]. The protein was set in a cuboid periodic box of 8 Å, between the protein and the periodic box wall, of TIP3P water molecules neutralized with Na+ and Cl- counter ions. The cut off for non-bonded interactions was set to 8 Å, employing particle-mesh Ewald summation with a fourth-order B-spline interpolation and a tolerance of $10^{-5}$. The non-bonded list was updated every 50 fs, and the MD time step was set to 2 fs, employing the SHAKE algorithm to constrain bonds involving hydrogen atoms.

The H atoms and residues #44, #46, #47 and #48 of the system were first minimized for 2000 cycles (1000 with steepest descendent and 1000 with conjugate gradient) by restraining the rest of the atoms with a 1000 kcal/mol/Å² force constant. The system is then minimized for another 3000 cycles (1000 with the steepest descendent and 2000 with conjugate gradient) restraining the non-hydrogen backbone atoms of the protein with a 10 kcal/mol/Å² force constant. Finally, the system was minimized for 10,000 cycles (2000 with the steepest descendent and 8000 with conjugate gradient), allowing all the atoms to relax.

The system was then heated from 0 to 300 K in the first 800 ps of an overall 1 ns run, using a NVT ensemble, employing Langevin dynamics with a collision frequency of 5 ps⁻¹. The system was then equilibrated for 1 ns in NPT ensemble at 300 K with isotropic position scaling and a relaxation time of 5 ps. The production phase was carried out using the same ensemble and parameters as in the equilibration phase. The production was performed for 150 ns for the SONOS and 600 ns for the reduced and disulfide-containing dimers. The MAH-covalent bound system was simulated for a total time of 3.5 µs. The fully crosslinked system (C145-MAH-K137) was simulated for a shorter time, given that large structural changes were observed in this time period. The heating and production phases were performed using graphics processing units (GPUs). The molecular dynamics runs used in our analysis were not repeated for statistical treatment. Instead, extended simulation times were used.

## Analysis of the molecular dynamics

All the structural analysis of the molecular dynamic simulations were performed using the CPPTRAJ (V4.25.6) tool from AmberTools (V20.15)[87]. The dimerization energies were analysed using the MMPBSA.py implementation[88] making use of 500 frames extracted from the first 150 ns of each system MD.

## Reporting summary

Further information on research design is available in the Nature Portfolio Reporting Summary linked to this article.

# Data availability

The refined structural protein models and corresponding structure-factor amplitudes have been deposited under PDB accession codes 7ZB6 (Mpro C44S), 7ZB7 (Mpro Y54F) and 7ZB8 (Mpro K61A). The structures cited in this publication are available under their respective PDB accession codes 6LU7, 7JR4 and 7KPH. Source data are provided for the kinetic analysis of enzyme activity (SARS-CoV-2 M^pro wild-type and variants, SARS-CoV M^pro wild-type) as well as all Western blots. Data of the quantum chemical calculations and MD simulations are provided in a public repository (https://doi.org/10.25625/GBIC2M). All other data are available on request. Source data are provided with this paper.

# Code availability

Codes used for the quantum chemical calculations and MD simulations are provided in a public repository (https://doi.org/10.25625/YMLHRB).

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

## Acknowledgements

This study was supported by the Max-Planck Society and the DFG-funded Göttingen Graduate Center for Neurosciences, Biophysics, and Molecular Biosciences GGNB (to K.T.). We further thank the Coronavirus Forschungsnetzwerk Niedersachsen (COFONI) for funding project 13FF22 to M.D. The analytical ultracentrifuge Beckman Coulter Optima AUC was funded by the Deutsche Forschungsgemeinschaft (DFG) – INST 192/534-1 FUGG (to U.C.). The study was also supported through DFG grants MA5063/4-1 (to R.A.M.) and 417677437/GRK2578 (to C.B.). The study was also supported by the German Center for Infection Research (DZIF) (to R.H.). We acknowledge access to beamline P14 at DESY/EMBL Hamburg, Germany and thank G. Bourenkov and T. Schneider for local support. We thank H. Sies for the helpful discussions. We thank R. Golbik for the discussion of the circular dichroism data and Lidia Litz for excellent technical assistance with the analytical ultracentrifugation experiments.

## Author contributions

K.T. conceptualized and coordinated the study. L.M.F., M.W., F.R.v.P., E.Pe. and N.E. recombinantly expressed all proteins and analyzed the enzymatic and biophysical properties under the supervision of K.T. G.P. and C.B. analyzed the redox modifications by redox proteomics. U.C. carried out and analyzed the analytical ultracentrifugation experiments. G.H. conducted the site-directed mutagenesis experiments. E.Pa. crystallized the variants C44S, Y54F and K61A. A.C. and A.R.P. collected datasets of protein crystals, and A.C. and F.R.v.P. refined the crystallographic structures. F.R.v.P. and K.T. interpreted the crystallographic data. J.U., T.F., S.B. carried out the quantum chemical calculations under the supervision of R.A.M. K.S. and A.D. carried out the cell culture experiments under the supervision of M.D. L.M.F., F.R.v.P., G.P., C.B., K.S., M.D., J.U., R.A.M., R.H., U.C. and K.T. wrote the paper with input from all the other authors.

## Funding

 **17**

## Competing interests

The authors declare no competing interests.
