## [Peer Review File · Nature Communications]

Multiple redox switches of the SARS-CoV-2 main protease in vitro provide new opportunities for drug designREVIEWER COMMENTS

Reviewer #1 (Remarks to the Author):

Overview: This is an interesting manuscript in which the authors demonstrate that Mpro can be inactivated with hydrogen peroxide and that this effect is primarily due to oxidation of the active site residue. They also provide evidence that certain compounds can crosslink the critical residues and inhibit SARS-CoV-2 infection.

Major Issues:

1. The authors state that Mpro reversibly switches between the enzymatically active dimer and the functionally dormant monomer through redox modifications of cysteine residues. The authors further claim to demonstrate a specific role for Cys117 in protecting the active site Cys145, based on the finding that they can reverse the activity of all individual Cys->Ser mutants of Mpro except for that of C117S. However, the other Cys->Ser mutants have only slight reversibility, ranging from 0.3% to 6%. One interpretation is that oxidation with H₂O₂ leads to irreversible oxidation of the majority of Cys145 (and activity) with or without Cys117. This data suggest that Cys117 is very weakly, if at all, protective. A more convincing case could be made if the authors were able to reverse significantly more of the activity of the WT enzyme with longer incubations with DTT, etc. In Supp Figure 6 the authors show that WT Mpro can be inactivated over a 20h period with H₂O₂ and that this is fully reversible. The authors should do the same experiment side by side with C117S and then determine the extent to which Cys117 is protecting Cys145.
2. The data from Table 1 suggests that Cys117 protects at best 6% of the activity when Mpro is oxidized by hydrogen peroxide. What about the rest of the activity? Also, it seems that if Cys117 is protecting Cys145 from becoming irreversibly oxidized then it would reason that in the absence of Cys117, oxidation would lead to an irreversible oxidation of Cys145 to the sulfonic acid derivative (sulfonylation). Again, the authors should compare WT vs C117S to try to demonstrate that in the absence of Cys117 there is increased level of Cys145 sulfonylation (or disulfides etc) thus demonstrating directly that 145 has become more susceptible to irreversible oxidation. Absent this data an alternate hypothesis is that when Cys117 is mutated to Ser then it leads to the irreversible oxidation of an alternate Cys residue and that this leads to inhibiting dimerization. Cys117 may be a primary target but when mutated, secondary targets may be hit. The authors should consider this point.
3. The authors make a more convincing case that oxidation of Mpro leads to partial monomerization of the enzyme which is reversible. This contrasts to the enzyme activity which is clearly only 6% reversible (see table 1) yet fully reversible in Supp Figure 6 (but only WT has been done). The authors state that the redox-dependent shift on the quaternary level is not fully reversible for C117S, while all other variants tested undergo a fully reversible switching. However, there are two issues. First, in Figure 2B (WT) the monomer is not fully reversed to dimer, and therefore authors should correct the statement to “mostly reversed”. Also, it is pointed out that C117S runs as a mixture of monomer and dimer even before oxidation, which makes interpretation of the role for C117 more difficult since mutation of C117 alone is affecting the K_d for dimerization. These points should be addressed.

4. Regarding the existence of a SONOS bridge at C22-K61-C44, more direct evidence for this could be obtained by the authors by determining the molecular mass of the WT Mproox before and after reduction and comparing the mass changes in C22S and C44S Mproox in the same case. Even more directly they could compare the WT to C22S and C44S double mutant. Further, in Figure 3B they mention that by MS they can detect disulfides of 117ss145, C117ssC300 and C145ssC300. The authors should consider the spontaneous deamidation of the Q terminal peptide to make sure the amount of oxidation they are seeing for the c-terminal peptide is accurate.
5. In Figure 6: The authors show MAH causes monomer formation in WT but have no data on other Mpro's. It would be very useful to see that C145S does not undergo this change to monomer thus implicating the requirement for C145. Also, data on the other mutants for the reactive residues in question could provide useful insight as well.
6. It is quite surprising that the drugs (especially BMOE) works against SARS-CoV-2 infection (Supp Fig 31) at about the same concentration at which it inhibits Mpro (Supp Figure 25b). This is substantially different from other studies, for example colloidal bismuth subcitrate (CBS) in Tao et al., J Chemical Science, 42, 2021, <https://doi.org/10.1039/D1SC03526F>. The authors should address this discrepancy. Also, the authors should provide a dose-response curve of the toxicity of the drugs, especially BMOE.

Other comments:

1. It is not clear what the buffer is when analyzing Mpro by gel filtration. Is it the same as that used for AUC (100 mM NaCl, Tris pH 7.3 +/-DTT)?
2. Virus/drug Data: This data would be more compelling if it were also done on a cell-based model that has an Mpro activity readout as this would then strengthen the antiviral data as being due, at least in part, to inhibition of Mpro. Without that, they have a weak argument for the compounds in vivo as they point out that these compounds can be rather promiscuous.
3. Finally, it would be of interest if the authors could consider what advantage it might be for the virus to have evolve so it its Mpro is affected so much by redox regulation.

Reviewer #2 (Remarks to the Author):

We have now reviewed the manuscript by Dr. Tittmann entitled "Multiple redox switches of the SARS-CoV-2 main protease in vitro provide new opportunities for drug design".

Previous work of the group identified in silico the protease Mpro from SARS-CoV-2 as a potential carrier of a NOS or SONOS bridge (a covalent crosslink between a C and a K residue, or among three C22-K61-C44 residues). With this manuscript, the group aims at characterizing the role of the bridge in enzyme activity and in protection from oxidation. Unfortunately, we do not find indications that this bridge is biologically relevant. A NOS or SONOS bridge has not been experimentally detected. Experimental evidence explaining the role of this hypothetical bridge on enzyme activity or activity preservation is not provided.

The in vitro H₂O₂ treatments use concentrations too high to be compared with those exerting toxicity in

vivo (in the low μM range). Based on their proposal, aerated buffers are sufficient to disrupt dimer conformation, but nevertheless the authors do not indicate whether this has an impact on protease activity, and to which extent.

A very important outcome of the manuscript is coming from the individual generation of C and K mutants (Fig. 3a and Table 1). The roles of C117 and C44 in enzyme activity are quite straightforward. The first one could be protecting the catalytic C145 from oxidation forming a disulfide, and whether this disulfide is formed with only aerated buffers is not shown.

Other comments:

1. As said above, the formation of monomers from dimers is connected to loss of activity (F1b, SI F8....). Formation of monomer can be observed in aerated, non-DTT containing buffers, similar to what is seen with H_2O_2 . However, the authors never show activity assays under this condition.
2. Analysis of C117S mutant is interesting. Why is it (almost) always a monomer (F2b)? The disulfide of C145 and C117 has been shown by mass spec, but if this disulfide protects the protein from oxidation, why does C117S have an impact on enzyme activity even with DTT in the buffers? Are other residues, such as C145, irreversibly modified in this mutant, as could be shown with mass spec?
3. C145S mutant: why is it (almost) always a dimer, similar to what is observed with 20 mM H_2O_2 in wild type Mpro?
4. As indicated in pag. 11, there is a manuscript (ref. 45) demonstrating that C300 glutathionylation during oxidative stress could break the Mpro dimer. How is this connecting to the redox switch proposed here?
5. Table 1: why only a small % of the enzyme activity (1-5%) is recovered in these assays upon DTT treatment, relative to the bars shown in SI Fig. 6 or Fig 5a (for SARS-CoV)? Is it due to the type of stress (100 μM or 1 mM H_2O_2), or the time of reduction with 20 mM DTT?
6. Table 1: most mutants lose a similar %, around 60%, of activity upon peroxide addition, even the mutant proteins lacking residues essential for the hypothetical double redox switch. Does it make sense that the reduction of enzyme activity is seen under reducing conditions for these mutants, but they do not behave worse than a wild-type during H_2O_2 -dependent inactivation?
7. Table 1: lack of effect of C22S mutations suggests that SONOS is not very important for activity.
8. Table 1 and other figs: C44S is clearly having an effect on activity even with DTT buffers. Since the SONOS nor the NOS bridge cannot be detected, could this be caused by the proximity to the active center (Y54)?
9. Table 1: K61 seems to lose 50% of the activity.
10. F3a: sulfenylation is not relevant upon 1 or 20 mM H_2O_2 . This is a very transient modification, which should be detected at low peroxide levels.
11. Pag. 7: it is claimed that NOS bridge had been proven before in reference 28. Do the authors refer to a different protein, not to Mpro? It was confusing in the text. Not even in the other reference by the group, which only proposes from sequence data the formation of the bridge.
12. We believe that 32 supp. figs were not really necessary

Reviewer #3 (Remarks to the Author):

The manuscript by Funk et al. described a thorough analysis of redox sensitive cysteines in the SARS-CoV-2 main protease and made exploitation of modifications of these cysteines for potential drug development. The paper started from the identification of cysteines that undergo redox modifications. Several key cysteines that undergo crosslink to form NOSOS and disulfide bounds are confirmed. By testing molecules that are bifunctional in modifying cysteine and lysine, two potent inhibitors were discovered. The work is a combination of experimental kinetic analysis, mutagenesis, crystallography, and molecular simulation. There are substantial data reported. Most experiments were designed and rigorously conducted. The conclusions are sound. This reviewer would like to support its acceptance after addressing some concerns.

Major:

1. The in vitro testing of oxidation of main protease used 100 uM or 1 mM H₂O₂. These conditions will not be likely happening physiologically even in cells undergoing inflammation (the authors may provide references to refute this claim). Even with this condition, 23 h for observing 50% activity loss in 100 uM H₂O₂ doesn't seem a detrimental effect to an enzyme. Based on the data presented, the enzyme is quite robust. Although the authors tried to rationalize the data by connecting to physiological conditions and assuming oxidation at 40 degrees will be 10 times fast, giving that the cells are quite reductive, it is not convincing that this will happen in cells. Actually, it is not sure whether the oxidation will be really 10 times fast at 40 degrees. However, whether this is physiological relevant doesn't decrease the quality of the study.
2. A similar C117-C145 disulfide crosslink was observed in the SARS-CoV main protease. Although the authors have quoted a previous publication about this, it is better to discuss it at the beginning. The two enzymes are highly homologous. It is very likely that the same thing happens for both.
3. MAH and BMOE are super-reactive molecules. It is not possible that they don't show toxicity. The observed EC₅₀ values are highly likely resulted from cell killing effects. A more robust cell toxicity test is to use 293T cells. 293T cells are not cancer cell lines. They are more sensitive to small molecule toxins. For most cancer cell lines, they tend to be resistant. Cells have high concentrations of cysteine and glutathione. They will neutralize MAH and BMOE. Selling them as possible antivirals makes no sense. However, the in vitro study itself is interesting and important. Just the part related to antivirals was oversold.

Minor:

1. Please update ref. 30. It has been published in ACS Chemical Biology.

Reviewer #4 (Remarks to the Author):

The manuscript investigates the mechanism underlying the redox-sensitivity of SARS-CoV-2 Mpro. The comprehensive analysis reveals the contributions of several key residues including C145, C117, C44, C22 and K61, as well as the inhibitory activity of compounds reacting with the thiol and amine groups. The results offer valuable insights into the redox-regulation of an important drug target, and will facilitate the testing and development of future inhibitors. The manuscript is well written and require only some minor revisions.

- 1) The mutant structures, especially those of C44S and Y54F, exhibited minimal changes compared with the WT. Can the authors compare the B factors of key active site residues (e.g. H41, C145, etc) to those of the corresponding WT structures (acyl intermediate or unbound), and see if there are any significant differences (vs the overall B factor of the whole structure)?
- 2) SI Fig.10c the acyl-enzyme bond in the mutant structure does not appear to be as planar as the WT. This may be partly due to the viewpoint. But please double check.
- 3) SI Table 1, the units for several parameters are missing, e.g., resolution, B factor, RMSD bond angles, Ramachandran statistics (%). The significant digits of 'resolution' should also be kept consistent among the three structures.
- 4) There are several typos or grammar mistakes, such as 'monemer' (monomer), 'unusual high' (unusually high), 'Its biological function...', 'establish' (establishes), 'physiological relevant' (physiologically), 'in case of MAH' (in the case of), etc.

Reviewer #5 (Remarks to the Author):

This paper presents convincing evidence for the conclusion that M_{pro} employs redox switches to regulate its function. This conclusion is supported by data from enzymatic assay; binding assays; mass spectrometry; crystallography, and molecular dynamics simulations. In vitro and in cell data are also presented to inhibitory effects of compounds that target the redox switches. I have several comments that I'd like the authors to address to further increase confidence in the work and improve readability.

1. p. 9: "Mass spectrometric analysis identified catalytic C145 and K137 as MAH-binding residues" -- the authors need to elaborate on the evidence for this interpretation, e.g., how unique is this interpretation? Are there alternative explanations or caveats? The same issue for the targets of the BMOE crosslinker.
2. Fig. 4: use of MMPBSA -- discussion of earlier studies using this method on M_{pro} (e.g., PMID 33119257) will add to the justification for MMPBSA.
3. The paper generally reads well, but I found a number of places where the language is awkward, incomplete, or confusing:

p. 3: "The flexibility of the loop is also confirmed by our molecular dynamics (MD) simulations (SI Fig. 3) in good agreement with temperature-dependent structural data of Mpro" -- it's unclear what exactly is the "good agreement".

p. 4: "eclipse period 10 h at 37-40 °C and when assuming a ~10-fold increased Mpro oxidation rate at ~40 °C relative to 0 °C according to Eyring theory" -- it appears the authors are arguing that the time determined on ice would be shortened by 10-fold if the measurement were made at 40 deg C. If so, explicitly state that, and also justify the 10-fold number.

p. 4: "protein concentration tested (0.25 μM) suggesting a KDapp of <250 nM" -- they should stick with either μM or nM, not switching units. The same issue for H2O2 concentration, it's stated as 100 μM in some places but as 0.1 mM in other places.

p. 4: "...spectroscopy under oxidizing and reducing conditions indicates small but reproducible structural differences between the two states" -- it should be noted that, in either state, the protein is a mixture of monomer and dimer.

p. 5: "In order to identify these cysteine residues, which are part of the redox switch(es)" -- awkward. How about "In order to identify cysteine residues that are part of the redox switch(es)"?

p. 5: "In the case of variant C44S, we were even able to obtain a structural snapshot of the covalent acyl intermediate" -- but the covalent acyl intermediate was already previously captured for WT (7KHP; as stated in SI Fig. 10 caption), so "even able" is uncalled for.

p. 5: "for practical reasons that is shorter reaction times" -- awkward and need to rephrase.

p. 7, second paragraph is particularly confusing regarding what H2O2 concentrations are needed and whether the products are sulfonylated or sulfenylated. The authors need to first explain the chemical differences b/w these forms to call attention to the small difference in spelling. Then present results for one form and then the second form, rather than mixing the results b/w the two forms.

p. 7: "variant C117S was the only variant" -- awkward. How about "C117S was the only variant"?

p. 8: "the reduced state indicates that residues C145 and C117 cannot directly form a disulfide bond as residue N28 lies in between the two side chains" -- confusing. How about "cannot form without significantly structural changes"?

p. 9: "Establishment of the second crosslink with K137 provided 12.0 kcal/mol" -- confusing. Maybe the authors mean that, with the second crosslink, the binding enthalpy increases to 12 kcal/mol?

SI Fig. 3 caption: "which are blocked or effectively block from solvent" -- awkward and unclear.

SI Fig. 26 caption: "MAH crosslinking is shown in Figure 4c of the main manuscript" -- they probably meant Fig. 6c?

Response to reviewers

On behalf of all coauthors, I would like to thank all reviewers for the constructive comments and thorough survey of our manuscript

“Multiple redox switches of the SARS-CoV-2 main protease in vitro provide new opportunities for drug design”

submitted as an article to *Nature Communications*.

In light of the reviewers' comments, we have revised our manuscript in numerous places and now include novel data as suggested by the referees. **Changes are indicated in yellow shade**. A point-by-point discussion is given below.

Reviewer 1

1.) The authors state that Mpro reversibly switches between the enzymatically active dimer and the functionally dormant monomer through redox modifications of cysteine residues. The authors further claim to demonstrate a specific role for Cys117 in protecting the active site Cys145, based on the finding that they can reverse the activity of all individual Cys->Ser mutants of Mpro except for that of C117S. However, the other Cys->Ser mutants have only slight reversibility, ranging from 0.3% to 6%. One interpretation is that oxidation with H₂O₂ leads to irreversible oxidation of the majority of Cys145 (and activity) with or without Cys117. This data suggest that Cys117 is very weakly, if at all, protective. A more convincing case could be made if the authors were able to reverse significantly more of the activity of the WT enzyme with longer incubations with DTT, etc. In Supp Figure 6 the authors show that WT Mpro can be inactivated over a 20h period with H₂O₂ and that this is fully reversible. The authors should do the same experiment side by side with C117S and then determine the extent to which Cys117 is protecting Cys145.

Authors:

We thank the referee for referring to that important point. Actually, we have conducted the requested experiments but they were 'hidden' in the vast amount of data. For testing reversibility of redox switching on the level of enzymatic activity, we did two types of experiments: one overnight for the wild-type and important variants (incl. C117S), and one for a few hundred seconds, where the wild-type already re-gains a few percent activity. The latter experiment was used as a proxy to identify variants with impaired reversibility. To make that point clearer, we have rewritten essential parts of that section. In sum, both types of experiments show full reversibility for the wild-type (new Fig. 2) and irreversibility for C117S. All other variants showed a similar re-activation in the proxy test compared to the wild-type. We have also updated Table 1 in this regard to make things clear.

p. 3/4

Redox-regulated enzymatic activity and oligomeric equilibria of M^{pro}

We had initially observed that M^{pro} loses enzymatic activity over a couple of days on ice when kept in non-reducing buffer (aerated buffer devoid of reductants such as e.g. DTT). Enzymatic activity could be fully restored when the enzyme was reacted with DTT. In order to test for and analyze a potential redox regulation of M^{pro} quantitatively, we subjected the protein to different levels of oxidative insult with H₂O₂ including a) 100 μM H₂O₂ as an upper limit for physiologically relevant oxidative stress conditions, b) 1 mM H₂O₂ or c) 20 mM H₂O₂ as a supraphysiological concentration^{35,36}.

We first measured the enzymatic activity under reducing versus oxidizing conditions and tested for reversibility of redox switching. The data are exemplary shown for the treatment with 100 μM H₂O₂ (**Figure 2**). We observed a progressive but essentially reversible loss of enzymatic activity over time that could be fully reversed upon treatment with the reductant DTT. When kept on ice, inactivation takes place over a time of 10-20 hours. This would be seemingly physiologically relevant in view of the SARS-CoV-2 replication time and reported eclipse period of 10 h at 37-40 °C³⁷ as the rate of (nonadiabatic) chemical reactions is typically 2-3fold higher per 10 K temperature increase. Using this approximation, the oxidation of M^{pro} by 100 μM H₂O₂ should be 8-12 time faster at 37-40 °C compared to 0 °C. The inactivation upon treatment with 1 mM H₂O₂ proceeds – as expected – faster, that is on a time scale of a few hours and is also fully reversible (**SI Fig 5a** and **Table 1**). At 20 mM H₂O₂, however, enzymatic activity is irreversibly lost (**SI Fig 5a**). This observation indicates that the protein becomes irreversibly overoxidized under these conditions, presumably through oxidation of the catalytic Cys145. At concentrations up to 1 mM H₂O₂, the catalytic cysteine Cys145 is well protected against overoxidation involving either sulfenic acid, a disulfide or a lysine-cysteine switch.

We further conducted gel filtration experiments with the reduced and oxidized protein (**SI Fig. 5b**). For practical reasons, we opted to use 1 mM H₂O₂ in case of oxidizing conditions as a) the oxidized protein can be obtained within a few hours at these H₂O₂ concentrations (see above) and b) switching is fully reversible. In the reduced state, M^{pro} is almost exclusively present as the dimer under the conditions used (only the dimer has enzymatic activity¹²). When treated with 1 mM H₂O₂, however, the M^{pro} dimer undergoes a dissociation leading to the formation of a marked monomer fraction. A re-reduction of oxidized M^{pro} leads to the quantitative formation of the dimer indicating a fully reversible redox switch on the oligomer level. Interestingly, when using supraphysiological concentrations of H₂O₂ – that is 20 mM – monomerization is not observed.

p.6:

We then measured the enzymatic activity under defined oxidizing conditions and tested whether a putative loss of activity is reversible. For the latter, we used the early onset of reactivation over the first 200 s after re-reduction with DTT as a proxy and analyzed variants with a kinetic phenotype in more detail with reduction taking place overnight. Treatment of M^{pro} wild-type and variants with 1 mM H₂O₂ for 2 h on ice results in decreased enzymatic activities to a similar extent in all proteins (~3-fold reduction) pinpointing the central role of catalytic residue C145 as a major site of redox modification (the only residue present in all tested variants) (**Table 1**). Re-reduction of the protein with DTT leads to a reactivation of enzymatic activity in all cysteine variants akin to the wild-type enzyme with the notable exception of variant C117S (**Fig. 3a, Table 1**). The activity of this variant is irreversibly lost upon oxidation even when reduction with DTT took place over night (**SI Fig. 11**). This is a clear indication that C117 is part of a redox switch involving C145, most likely in the form of a disulfide-dithiol switch. Interestingly, a C117-C145 disulfide was reported for a SARS-CoV M^{pro} variant but not the wild-type protein³⁴. In variant C117S, where no C145-C117 disulfide can be formed, C145 might not be protected against overoxidation and thus explain the irreversible nature of redox switching. For some of the SONOS variants such as triple variant C22S_C44S_K61A, only a partial recovery of enzymatic activity can be observed but only after long incubation times with reductant showcasing the structural importance of the SONOS motif for the correct functioning of the redox switch(es) of M^{pro} (**Table 1, SI Fig. 12**).

2.) The data from Table 1 suggests that Cys117 protects at best 6% of the activity when M^{pro} is oxidized by hydrogen peroxide. What about the rest of the activity? Also, it seems that if Cys117 is protecting Cys145 from becoming irreversibly oxidized then it would reason that in the absence of Cys117, oxidation would lead to an irreversible oxidation of Cys145 to the sulfonic acid derivative (sulfonylation). Again, the authors should compare WT vs C117S to try to demonstrate that in the

absence of Cys117 there is increased level of Cys145 sulfonylation (or disulfides etc) thus demonstrating directly that 145 has become more susceptible to irreversible oxidation. Absent this data an alternate hypothesis is that when Cys117 is mutated to Ser then it leads to the irreversible oxidation of an alternate Cys residue and that this leads to inhibiting dimerization. Cys117 may be a primary target but when mutated, secondary targets may be hit. The authors should consider this point.

Authors: see above. In addition, we have now conducted a head-to-head redox proteomics analysis of C117S and wild-type M^{pro} that shows that in C117S, catalytic C145 becomes more easily oxidized but also forms alternative disulfide bridges that seem to be kinetically stable. Overall, C117S is not as robust against oxidation as the wild-type enzyme.

p.9

To obtain insights into why variant C117S, in which the C117-C145 disulfide-dithiol switch has been defunctionalized, is not reversibly switching as the wild-type protein, we conducted a head-to-head redox proteomics analysis of C117S versus wild-type M^{pro}. First, we analyzed the oxidation states of catalytic residue C145 under different oxidizing conditions (SI Fig. 21). As expected, overoxidation (sulfonylation) of catalytic C145 in C117S is increased, but not to an extent that can fully explain the irreversible nature of redox switching in the variant. We therefore analyzed other modifications including disulfide bridges (SI Fig. 22). As discussed above, wild-type M^{pro} forms the C117-C145 disulfide as the major linkage. In C117S, however, a marked increase of the C145-C300 disulfide linkage is observed relative to the wild-type. Owing to the relatively close spatial proximity of C300 of one chain of the M^{pro} dimer to the catalytic site of the neighboring chain (see SI Fig. 1), this disulfide bridge is very likely an interchain crosslink. This finding would explain, why oxidation of variant C117S leads to formation of the dimer (see Fig. 3b). The irreversible nature of the redox switching in C117S would require that the formed C145-C300 disulfide is shielded from the solvent such that reductants as DTT cannot directly react thus constituting a kinetically stable disulfide under the conditions used. Overall, variant C117S is slightly more susceptible to oxidation at rather low H₂O₂ concentrations of 20 and 100 μM than the wild-type protein using the Western blot analysis of sulfonylation as a readout (SI Fig. 23).

3.) The authors make a more convincing case that oxidation of M^{pro} leads to partial monomerization of the enzyme which is reversible. This contrasts to the enzyme activity which is clearly only 6% reversible (see table 1) yet fully reversible in Supp Figure 6 (but only WT has been

done). The authors state that the redox-dependent shift on the quaternary level is not fully reversible for C117S, while all other variants tested undergo a fully reversible switching. However, there are two issues. First, in Figure 2B (WT) the monomer is not fully reversed to dimer, and therefore authors should correct the statement to “mostly reversed”. Also, it is pointed out that C117S runs as a mixture of monomer and dimer even before oxidation, which makes interpretation of the role for C117 more difficult since mutation of C117 alone is affecting the K_d for dimerization. These points should be addressed.

Authors: Actually, the wild-type protein undergoes quantitative dimerization with 1 mM H₂O₂ but not with 20 mM H₂O₂ the latter being a supraphysiological concentration. We have rewritten that section to convey the results more clearly (see point 1).

4.) Regarding the existence of a SONOS bridge at C22-K61-C44, more direct evidence for this could be obtained by the authors by determining the molecular mass of the WT M^{pro} before and after reduction and comparing the mass changes in C22S and C44S M^{pro} in the same case. Even more directly they could compare the WT to C22S and C44S double mutant. Further, in Figure 3B they mention that by MS they can detect disulfides of 117ss145, C117ssC300 and C145ssC300. The authors should consider the spontaneous deamidation of the Q terminal peptide to make sure the amount of oxidation they are seeing for the c-terminal peptide is accurate.

Authors: The NOS and SONOS bridges in M^{pro} could be recently directly detected by mass spec in a publication by Liu et al. In K61 mutant proteins, the SONOS bridge is absent. These authors analyzed the undigested protein as opposed to us. We refer to this in the redox proteomics section:

p. 8

The existence of SONOS-linked peptides (C22-K61-C44) could not be directly **proven by mass spectrometry of the proteolytically digested protein similar to the initially discovered NOS crosslink in a transaldolase²⁸**, but we noticed that residues C22 and C44 were only accessible for alkylation after reduction (following an initial oxidation with H₂O₂) implicating a previous oxidized state of both sites. As C22 and C44 were neither found to be sulfenylated, sulfinylated, sulfonylated (traces of sulfonylated C22 were found at supraphysiological H₂O₂ concentrations) nor in a disulfide linkage, this might be considered as indirect evidence for the existence of the SONOS bridge protecting those residues from getting further oxidized at physiologically relevant H₂O₂ concentrations. **Mass spectrometric evidence for the existence of the SONOS bridge was recently provided by Liu and colleagues who analyzed the undigested protein³⁰.**

Further, a deamidation can be excluded based on our data.

5.) In Figure 6: The authors show MAH causes monomer formation in WT but have no data on other Mpro's. It would be very useful to see that C145S does not undergo this change to monomer thus implicating the requirement for C145. Also, data on the other mutants for the reactive residues in question could provide useful insight as well.

Authors: In fact, we show that MAH also acts on M^{pro} from the original SARS-CoV (new Fig. 6). We have also started to systematically test different variants of M^{pro} from SARS-CoV-2 but believe that this goes beyond the scope of the current manuscript.

6.) It is quite surprising that the drugs (especially BMOE) works against SARS-CoV-2 infection (Supp Fig 31) at about the same concentration at which it inhibits Mpro (Supp Figure 25b). This is substantially different from other studies, for example colloidal bismuth subcitrate (CBS) in Tao et al., J Chemical Science, 42, 2021, <https://doi.org/10.1039/D1SC03526F>. The authors should address this discrepancy. Also, the authors should provide a dose-response curve of the toxicity of the drugs, especially BMOE.

Authors: We have been conducting the experiments as suggested by the reviewer. We observe that there is some mild cytotoxicity of MAH and BMOE in the concentration regime, where antiviral activity is seen.

p. We could rule out that the assigned antiviral activity of MAH and BMOE results from their cytotoxicity (SI Fig. 34). Both compounds exhibit a mild cytotoxic effect in the concentration regime relevant for the dose:response analysis, in particular BMOE, but become clearly cytotoxic at higher concentrations. We would like to note, however, that cytotoxicity of both compounds might be more pronounced in other cell lines.

SI Figure 34. Analysis of the cytotoxic effect of crosslinkers MAH (panel a) and BMOE (panel b) on Vero E6 cells using the LDH assay (see Materials and Methods section). Note the mild cytotoxic effect of MAH up to concentrations of 40 mM and the slightly stronger cytotoxicity of BMOE in the same concentration regime. Since the inhibition profiles of both crosslinkers in the dose response experiments are in a regime of little cytotoxicity (see **SI Figure 33**), we conclude that the inhibitory effect mostly results from impairing SARS-CoV-2 propagation.

Other comments

1. It is not clear what the buffer is when analyzing Mpro by gel filtration. Is it the same as that used for AUC (100 mM NaCl, Tris pH 7.3 +/-DTT)?

Authors: Yes, this is correct. We have added the info in the *Material and Methods* section.

2. Virus/drug Data: This data would be more compelling if it were also done on a cell-based model that has an Mpro activity readout as this would then strengthen the antiviral data as being due, at least in part, to inhibition of Mpro. Without that, they have a weak argument for the compounds in vivo as they point out that these compounds can be rather promiscuous.

Authors: We do agree and are aware of this fact. We believe that we fairly discuss this and provide points as to why virus inhibition is nonetheless a likely scenario.

p.11

Quantitative analysis of the dose response curves (virus RNA progeny) shows that the estimated EC₅₀ values are in the millimolar range and thus higher than the IC₅₀ values obtained under in vitro

conditions with highly enriched M^{pro} in buffer (**SI Fig. 33**). The EC₅₀ value for BMOE is ~10 times smaller (0.8 mM) than that of MAH (7.9 mM) akin to the in vitro studies and estimated IC₅₀ values. The almost identical ratios for EC₅₀ ($EC_{50}^{BMOE}/EC_{50}^{MAH}$) and IC₅₀ values ($IC_{50}^{BMOE}/IC_{50}^{MAH}$) would seemingly suggest that antiviral activity of these compounds is reflecting inhibition of M^{pro}. Admittedly, we cannot rule out that these compounds inhibit viral replication by “off-target” effects on other viral proteins, e.g. the papain-like protease, or host proteins.

3. Finally, it would be of interest if the authors could consider what advantage it might be for the virus to have evolve so it its M^{pro} is affected so much by redox regulation.

Authors: Viral infections are known to be accompanied by oxidative stress. As M^{pro} is a cysteine protease it is particularly vulnerable against oxidative stress as the catalytic cysteine might become overoxidized leading to an enzymatically inactive protein. We refer to this at different places of the manuscript.

p. 3

The involvement of catalytic cysteine residues is a potential Achilles heel for viral replication, as oxidative stress exerted by the host innate immune system in response to viral infection may irreversibly (over)oxidize the cysteines and thus inactivate the enzyme and block replication^{25,26}. Although M^{pro} resides in the cytoplasm, which is typically considered to be of reducing nature, it has been established that oxidative bursts or even physiological redox signaling based on enzymatic production of reactive oxygen species (ROS) such as H₂O₂ leads to local oxidizing conditions and subsequent oxidation of protein thiols in the cytosol.

p. 12

In summary, we have reported on a hitherto unknown mode of redox regulation of SARS-CoV-2 M^{pro} in vitro that protects the redox-vulnerable catalytic cysteine and the structural integrity of the protein under oxidative stress conditions that are known to accompany SARS-CoV-2 infection^{43,44}. The in vivo relevance of the redox switches remains to be confirmed (or ruled out) in future studies, yet it seems that the detected time scales of the underlying processes and oxidation conditions are in principle compatible with physiological oxidative stress conditions. The unusual high abundance of cysteine residues (4%) distributed over the protein molecule would ensure resistance to oxidative stress conditions not only by scavenging of ROS but by multiple sophisticated redox switches that protect against overoxidation of the catalytic cysteine (disulfide-dithiol with C117) and against destabilization of the three-dimensional protein structure by

establishing a trivalent SONOS bridge that tethers three spatially proximal structural units via residues C22, C44 and K61.

Reviewer 2

Main 1.) We have now reviewed the manuscript by Dr. Tittmann entitled “Multiple redox switches of the SARS-CoV-2 main protease in vitro provide new opportunities for drug design”.

Previous work of the group identified in silico the protease M^{pro} from SARS-CoV-2 as a potential carrier of a NOS or SONOS bridge (a covalent crosslink between a C and a K residue, or among three C22-K61-C44 residues). With this manuscript, the group aims at characterizing the role of the bridge in enzyme activity and in protection from oxidation. Unfortunately, we do not find indications that this bridge is biologically relevant. A NOS or SONOS bridge has not been experimentally detected. Experimental evidence explaining the role of this hypothetical bridge on enzyme activity or activity preservation is not provided

Authors: The NOS and SONOS bridges could be directly detected by mass spec by our peers Liu et al.. We have added this information to the manuscript (see our response to point 4, reviewer1). Also, we would like to kindly disagree that there is no function of the SONOS bridge. The mutant data clearly show that variants with mutations of the SONOS residues, in particular these with exchanges of K61, are prone to unfolding/aggregation under oxidizing conditions (please see SI Fig. 15c). We refer to this in the text.

p.7

SONOS variants, in particular those containing an exchange of **residue** K61, are very susceptible to aggregation **under oxidizing conditions** and exhibit an atypical early onset of thermal denaturation (30-35 °C) with almost no cooperativity of unfolding, indicating a very loosely structured protein (**SI Fig. 15c**). **Under reducing conditions, the protein is structurally stable**. This would imply a structurally stabilizing function of the SONOS bridge under oxidizing conditions. The X-ray crystallographic analysis of the K61A variant in complex with the acyl intermediate formed between C145 and Q306 of a symmetry-related M^{pro} molecule indeed reveals **marked** structural changes throughout the whole molecule with an r.m.s.d. of the C α -carbons of 2.87 Å for chain A and 3.10 Å for chain B, respectively, compared to the wild-type structure (**SI Fig. 16**).

p 9.

Overall, the redox switch mechanism protects M^{pro} against oxidative damage by forming a C145-C117 disulfide as a stable storage of the catalytic cysteine avoiding an irreversible overoxidation

or formation of other kinetically stable modifications. The SONOS bridge structurally stabilizes the monomer that is formed upon oxidation by covalently tethering three structural elements within a monomer.

Main 2.) The in vitro H₂O₂ treatments use concentrations too high to be compared with those exerting toxicity in vivo (in the low uM range). Based on their proposal, aerated buffers are sufficient to disrupt dimer conformation, but nevertheless the authors do not indicate whether this has an impact on protease activity, and to which extend.

Authors: we have added the information (see our answer to point 1 of reviewer 1).

1. As said above, the formation of monomers from dimers is connected to loss of activity (F1b, SI F8...). Formation of monomer can be observed in aerated, non-DTT containing buffers, similar to what is seen with H₂O₂. However, the authors never show activity assays under this condition.

Authors: we have indeed observed a loss of activity, when M^{pro} was stored in nonreducing buffers. This was accompanied by formation of a monomer fraction in the gel filtration experiments.

See p 3.

We had initially observed that M^{pro} loses enzymatic activity over a couple of days on ice when kept in non-reducing buffer (aerated buffer devoid of reductants such as e.g. DTT). Enzymatic activity could be fully restored when the enzyme was reacted with DTT. In order to test for and analyze a potential redox regulation of M^{pro} quantitatively, we subjected the protein to different levels of oxidative insult with H₂O₂ including a) 100 μM H₂O₂ as an upper limit for physiologically relevant oxidative stress conditions, b) 1 mM H₂O₂ or c) 20 mM H₂O₂ as a supraphysiological concentration^{35,36}.

2.) Analysis of C117S mutant is interesting. Why is it (almost) always a monomer (F2b)? The disulfide of C145 and C117 has been shown by mass spec, but if this disulfide protects the protein from oxidation, why does C117S have an impact on enzyme activity even with DTT in the buffers? Are other residues, such as C145, irreversibly modified in this mutant, as could be shown with mass spec?

Authors: The mutation leads to a change of the protein structure as it is part of the redox switch machinery. We now comment on this in the manuscript.

p. 6

Variant C117S was unique in forming a detectable fraction of the monomer already under reducing conditions. Intriguingly, upon oxidation, the oligomeric equilibrium shifted to the dimer as opposed to the wild-type protein that shifts to the monomer. This indicates that the mutation of C117 leads to a structural rearrangement that is not confined to the local environment of C117 but also changes the structure of the dimer interface. Notably, the redox-dependent shift on the quaternary level is not reversible for variant C117S, while all other variants tested undergo a fully reversible switching. This observation suggests that oxidation of C117S leads to an irreversibly modified protein, presumably with an overoxidized catalytic C145.

3.) C145S mutant: why is it (almost) always a dimer, similar to what is observed with 20 mM H₂O₂ in wild type Mpro?

Authors: As the main redox switch (C117-C145 disulfide-dithiol) is absent in this variant, it cannot be switched to the monomer upon oxidation and is a dimer under both reducing and oxidizing conditions.

4.) As indicated in pag. 11, there is a manuscript (ref. 45) demonstrating that C300 glutathionylation during oxidative stress could break the Mpro dimer. How is this connecting to the redox switch proposed here?

Authors: We have added the following interpretation in the manuscript.

p. 12

Additional redox regulatory mechanisms might involve glutathionylation of cysteines (C300) as recently reported⁴⁵. Glutathionylation and disulfide formation might function synergistically as both drive dissociation of the functional dimer into the nonfunctional monomer. Also, a glutathionylation of C300 as reported might prevent formation of a kinetically stable C145-C300 disulfide crosslink.

5.) Table 1: why only a small % of the enzyme activity (1-5%) is recovered in these assays upon DTT treatment, relative to the bars shown in SI Fig. 6 or Fig 5a (for SARS-CoV)? Is it due to the type of stress (100 μ M or 1 mM H₂O₂), or the time of reduction with 20 mM DTT?

Authors: As pointed out in our response to point 1 of reviewer 1, we have conducted two different experiments to test for reversibility. One with a relatively short re-activation time (a few hundred seconds) to principally identify variants with impaired reactivation and second, overnight reactivation, in which full reversibility is detected for the wild-type.

6.) Table 1: most mutants lose a similar %, around 60%, of activity upon peroxide addition, even the mutant proteins lacking residues essential for the hypothetical double redox switch. Does it make sense that the reduction of enzyme activity is seen under reducing conditions for these mutants, but they do not behave worse than a wild-type during H₂O₂-dependent inactivation?

Authors: That critically depends on the relative ratio of the microscopic rate constants for the initial oxidation of C145 (sulfenic acid) versus that of disulfide formation (C117-C145) or further oxidation of mono-oxidized C145 to sulfinic and sulfonic acid. If the initial oxidation of C145 is overall rate-limiting (and disulfide formation or overoxidation occur way faster) then a similar reduction of activity will be seen for all variants.

7.) Table 1: lack of effect of C22S mutations suggests that SONOS is not very important for activity.

Authors: We would like to kindly disagree. The formation of the SONOS bridge occurs in stepwise manner. Both our data mining in the pdb as well as the work of our peers Liu et al. have shown that C22 and K61 form a NOS before C44 is engaged to form the SONOS. The C22-K61 is the platform for recruiting C44 (see our original publication in Nat. Chem. Bio. and SI Fig. 2).

8.) Table 1 and other figs: C44S is clearly having an effect on activity even with DTT buffers. Since the SONOS nor the NOS bridge cannot be detected, could this be caused by the proximity to the active center (Y54)?

Authors: We agree. We have stated in the manuscript:

p. 5

In addition, we produced single, double and triple mutants with individual and combined exchanges of the SONOS bridge residues C22, C44 and K61 including residue Y54 that directly interacts with C44.

Interestingly, variant Y54F, in which the tyrosine that interacts with SONOS residue C44 is replaced, shows a similar catalytic deficiency (19%) as variant C44S suggesting that both residues are required for full catalytic competence of the active site.

9.) F3a: sulfenylation is not relevant upon 1 or 20 mM H₂O₂. This is a very transient modification, which should be detected at low peroxide levels.

Authors: We agree with the reviewers. We have added novel data for the wild-type protein and variant C117S where sulfenylation was tested at 20 μM and 100 μM H₂O₂ (see new **SI Fig. 23**).

10.) Pag. 7: it is claimed that NOS bridge had been proven before in reference 28. Do the authors refer to a different protein, no to M^{pro}? It was confusing in the text. Not even in the other reference by the group, which only proposes from sequence data the formation of the bridge.

Authors: This was indeed SARS-CoV-2 M^{pro}. We analyzed the structural data (models and structure factors) in the pdb and found evidence for NOS and SONOS bridges based on the electron density maps. We have rewritten this section to convey the point more clearly.

p. 3

Interestingly, M^{pro} is amongst this class of proteins suggesting the possibility that it is redox regulated²⁹. Specifically, an allosteric SONOS bridge consisting of two cysteines (C22, C44) and one lysine (K61) within one protein chain was detected **by our mining of the protein data base and independent structural studies of Liu and coworkers** (**SI Fig. 2**)^{29,30}.

11.) We believe that 32 supp. figs were not really necessary.

Authors: We are aware that this study comprises a large dataset. As COVID and M^{pro} bear high relevance for human health, we have decided to show all primary data for assuring transparency.

Reviewer 3

Main

The manuscript by Funk et al. described a thorough analysis of redox sensitive cysteines in the SARS-CoV-2 main protease and made exploitation of modifications of these cysteines for potential drug development. The paper started from the identification of cysteines that undergo redox modifications. Several key cysteines that undergo crosslink to form NOSOS and disulfide bounds

are confirmed. By testing molecules that are bifunctional in modifying cysteine and lysine, two potent inhibitors were discovered. The work is a combination of experimental kinetic analysis, mutagenesis, crystallography, and molecular simulation. There are substantial data reported. Most experiments were designed and rigorously conducted. The conclusions are sound. This reviewer would like to support its acceptance after addressing some concerns.

Authors: We would like to thank this referee for the positive assessment of our study.

1.) The in vitro testing of oxidation of main protease used 100 μM or 1 mM H_2O_2 . These conditions will not be likely happening physiologically even in cells undergoing inflammation (the authors may provide references to refute this claim). Even with this condition, 23 h for observing 50% activity loss in 100 μM H_2O_2 doesn't seem a detrimental effect to an enzyme. Based on the data presented, the enzyme is quite robust. Although the authors tried to rationalize the data by connecting to physiological conditions and assuming oxidation at 40 degrees will be 10 times fast, giving that the cells are quite reductive, it is not convincing that this will happen in cells. Actually, it is not sure whether the oxidation will be really 10 times fast at 40 degrees. However, whether this is physiological relevant doesn't decrease the quality of the study.

Authors: We have rewritten the section relevant for this statement.

p. 4

We first measured the enzymatic activity under reducing versus oxidizing conditions and tested for reversibility of redox switching. The data are exemplary shown for the treatment with 100 μM H_2O_2 (**Figure 2**). We observed a progressive but essentially reversible loss of enzymatic activity over time that could be fully reversed upon treatment with reductant DTT. When kept on ice, inactivation takes place over a time of 10-20 hours. This would be seemingly physiologically relevant in view of the SARS-CoV-2 replication time and reported eclipse period of 10 h at 37-40 $^\circ\text{C}$ ³⁷ as the rate of (nonadiabatic) chemical reactions is typically 2-3fold higher per 10 K temperature increase. Using this approximation, the oxidation of M^{PrO} by 100 μM H_2O_2 should be 8-12 time faster at 37-40 $^\circ\text{C}$ compared to 0 $^\circ\text{C}$. The inactivation upon treatment with 1 mM H_2O_2 proceeds - as expected – faster, that is on a time scale of a few hours and is also fully reversible (**SI Fig 5a** and **Table 1**). At 20 mM H_2O_2 , however, enzymatic activity is irreversibly lost (**SI Fig 5a**). This observation indicates that the protein becomes irreversibly overoxidized under these conditions, presumably through oxidation of the catalytic Cys145. At concentrations up to 1 mM H_2O_2 , the catalytic cysteine Cys145 is well protected against overoxidation involving either sulfenic acid, a disulfide or a lysine-cysteine switch.

2.) A similar C117-C145 disulfide crosslink was observed in the SARS-CoV main protease. Although the authors have quoted a previous publication about this, it is better to discuss it at the beginning. The two enzymes are highly homologous. It is very likely that the same thing happens for both.

Authors: We agree with the referee and have now added this information in the introductory part and at several places in the results section.

p. 3

In case of the closely related M^{pro} from SARS-CoV, a disulfide modification between the catalytic C145 and C117 had been reported for a variant, in which residue N28 had been replaced ³⁴.

p. 6

Re-reduction of the protein with DTT leads to a reactivation of enzymatic activity in all cysteine variants akin to the wild-type enzyme with the notable exception of variant C117S (**Fig. 3a, Table 1**). The activity of this variant is irreversibly lost upon oxidation even when reduction with DTT took place over night (**SI Fig. 11**). This is a clear indication that C117 is part of a redox switch involving C145, most likely in the form of a disulfide-dithiol switch. Interestingly, a C117-C145 disulfide was reported for a SARS-CoV M^{pro} variant but not the wild-type protein ³⁴.

3.) MAH and BMOE are super-reactive molecules. It is not possible that they don't show toxicity. The observed EC50 values are highly likely resulted from cell killing effects. A more robust cell toxicity test is to use 293T cells. 293T cells are not cancer cell lines. They are more sensitive to small molecule toxins. For most cancer cell lines, they tend to be resistant. Cells have high concentrations of cysteine and glutathione. They will neutralize MAH and BMOE. Selling them as possible antivirals makes no sense. However, the in vitro study itself is interesting and important. Just the part related to antivirals was oversold.

Authors: We have added the mentioned caveat directly in the text and have added the toxicity data for both MAH and BMOE. It was clearly not our intention to sell both crosslinkers as lead compounds but to provide a proof-of-principle that compounds with two covalent warheads might switch M^{pro} into the inactive monomer.

p. 11

We could rule out that the assigned antiviral activity of MAH and BMOE results from their cytotoxicity (SI Fig. 34). Both compounds exhibit a mild cytotoxic effect in the concentration regime relevant for the dose:response analysis, in particular BMOE, but become clearly cytotoxic at higher concentrations. We would like to note, however, that cytotoxicity of both compounds might be more pronounced in other cell lines.

p.11

The finding of EC₅₀ values for BMOE and MAH in the mM range is not surprising given the high reactivity and lack of selectivity of the thiol and amine warheads of MAH (maleimide, NHS ester) and BMOE (maleimide). Nitril or ketoamide warheads as used for e.g. nirmatrelvir or other promising M^{pro}-targeting inhibitors are likely to be more selective, less toxic and chemically less reactive warheads binding to cysteine residues^{12,23}. While the amine function of lysine residues is typically considered a non-optimal target group for covalent drugs, very recent studies highlighted boronates (o-aminomethyl phenylboronic acid) or salicylaldehydes as promising compounds that reversibly bind to lysines in covalent fashion⁴¹⁻⁴³. We envisage the design of a covalent M^{pro} inhibitor with multiple covalently binding warheads as a promising direction to increase selectivity and efficacy using the redox switch mechanism described here as a blueprint. These could target the catalytic cysteine C145 and C117, which form the disulfide-dithiol switch, and/or K137 as these residues are conserved in many Coronavirus main proteases (SI Fig. 35). As proof-of-concept, we found that MAH inhibits M^{pro} from SARS-CoV in vitro in the same way as M^{pro} from SARS-CoV-2 leading to a destabilization of the dimer (Fig. 5c). Alternatively, SONOS residue C44 or glutathionylation site C300⁴⁵ at the dimer interface, both proximal to catalytic C145, could be target sites for covalently binding warheads.

Minor

Please update ref. 30. It has been published in ACS Chemical Biology.

Authors: We now refer to this study in the introduction and various other places in the manuscript.

Reviewer 4

Main: The manuscript investigates the mechanism underlying the redox-sensitivity of SARS-CoV-2 M^{pro}. The comprehensive analysis reveals the contributions of several key residues including C145, C117, C44, C22 and K61, as well as the inhibitory activity of compounds reacting with the thiol and amine groups. The results offer valuable insights into the redox-regulation of an important drug target, and will facilitate the testing and development of future inhibitors. The manuscript is well written and require only some minor revisions.

Authors: We would like to thank this reviewer for the positive assessment of our work.

1.) The mutant structures, especially those of C44S and Y54F, exhibited minimal changes compared with the WT. Can the authors compare the B factors of key active site residues (e.g. H41, C145, etc) to those of the corresponding WT structures (acyl intermediate or unbound), and see if there are any significant differences (vs the overall B factor of the whole structure)?

Authors: Thanks for making this point. We have added information in SI Fig 9 and 10.

SI Fig. 9. Please note that loop 43-50 bearing the mutation site exhibits increased flexibility indicated by elevated B-factors.

SI Fig. 10. Note the slight structural changes of catalytic residue H41 and its increased flexibility indicated by two alternate conformations.

2.) SI Fig.10c the acyl-enzyme bond in the mutant structure does not appear to be as planar as the WT. This may be partly due to the viewpoint. But please double check.

Authors: We have refined the conjugate with in-plane restraints with some slack, there is a small deviation from perfect planarity. Thanks for alluding to this.

3.) SI Table 1, the units for several parameters are missing, e.g., resolution, B factor, RMSD bond angles, Ramachandran statistics (%). The significant digits of 'resolution' should also be kept consistent among the three structures.

Authors: We have corrected the table in both aspects mentioned.

4.) There are several typos or grammar mistakes, such as 'monemer' (monomer), 'unusual high' (unusually high), 'Its biological function..., establish' (establishes), 'physiological relevant' (physiologically), 'in case of MAH' (in the case of), etc.

Authors: We double checked the manuscript now and hope to have eliminated these errors.

Reviewer 5

Main: This paper presents convincing evidence for the conclusion that M_{pro} employs redox switches to regulate its function. This conclusion is supported by data from enzymatic assay; binding assays; mass spectrometry; crystallography, and molecular dynamics simulations. In vitro and in cell data are also presented to inhibitory effects of compounds that target the redox

switches. I have several comments that I'd like the authors to address to further increase confidence in the work and improve readability.

Authors: We would like to thank the reviewer for the positive assessment of our work.

1.) 1. p. 9: "Mass spectrometric analysis identified catalytic C145 and K137 as MAH-binding residues" -- the authors need to elaborate on the evidence for this interpretation, e.g., how unique is this interpretation? Are there alternative explanations or caveats? The same issue for the targets of the BMOE crosslinker.

Authors: We could unambiguously identify the crosslinked peptides by mass spectrometric analysis (MS/MS). This data and the MD simulations with crosslinked protein (leading to monomerization as observed in the biochemical experiments) provide clear evidence for the proposed mode of action of the crosslinkers.

2.) Fig. 4: use of MMPBSA -- discussion of earlier studies using this method on Mpro (e.g., PMID 33119257) will add to the justification for MMPBSA.

Authors: we have been adding this reference to the main manuscript.

3.) language

a) p. 3: "The flexibility of the loop is also confirmed by our molecular dynamics (MD) simulations (SI Fig. 3) in good agreement with temperature-dependent structural data of Mpro" -- it's unclear what exactly is the "good agreement".

Authors: we have rewritten this section:

The loop is structurally highly flexible as revealed by our molecular dynamics (MD) simulations (SI Fig. 3) and also reported by temperature-dependent structure analysis of M^{pro} ³¹.

b) p. 4: "eclipse period 10 h at 37-40 °C and when assuming a ~10-fold increased Mpro oxidation rate at ~40 °C relative to 0 °C according to Eyring theory" -- it appears the authors are arguing that the time determined on ice would be shortened by 10-fold if the measurement were made at 40 deg C. If so, explicitly state that, and also justify the 10-fold number.

Authors: We have rewritten this section to convey the information in a better way.

This would be seemingly physiologically relevant in view of the SARS-CoV-2 replication time and reported eclipse period of 10 h at 37-40 °C ³⁷ as the rate of (nonadiabatic) chemical reactions is

typically 2-3fold higher per 10 K temperature increase. Using this approximation, the oxidation of M^{pro} by 100 μ M H₂O₂ should be 8-12 time faster at 37-40 °C compared to 0 °C. The inactivation upon treatment with 1 mM H₂O₂ proceeds - as expected – faster, that is on a time scale of a few hours and is also fully reversible (**SI Fig 5a and Table 1**).

c) "protein concentration tested (0.25 μ M) suggesting a KDapp of <250 nM" -- they should stick with either μ M or nM, not switching units. The same issue for H₂O₂ concentration, it's stated as 100 μ M in some places but as 0.1 mM in other places.

Authors: corrected

d) p. 4: "...spectroscopy under oxidizing and reducing conditions indicates small but reproducible structural differences between the two states" -- it should be noted that, in either state, the protein is a mixture of monomer and dimer.

Authors: We are aware of this and have rephrased the sentence in question:

Structure analysis of M^{pro} by far-UV circular dichroism (CD) spectroscopy under oxidizing and reducing conditions indicates small but reproducible structural differences between the two states based on secondary structure content (**Fig. 1c**).

e) "In order to identify these cysteine residues, which are part of the redox switch(es)" -- awkward. How about "In order to identify cysteine residues that are part of the redox switch(es)"?

Authors: Changed accordingly.

f) "In the case of variant C44S, we were even able to obtain a structural snapshot of the covalent acyl intermediate" -- but the covalent acyl intermediate was already previously captured for WT (7KHP; as stated in SI Fig. 10 caption), so "even able" is uncalled for.

Authors: We deleted "even".

g) p. 5: "for practical reasons that is shorter reaction times" -- awkward and need to rephrase.

Authors: The sentence has been rewritten:

For practical reasons, we used 1 mM H₂O₂ for oxidizing conditions as wild-type M^{Pro} undergoes a fully reversible redox switching under these conditions and oxidized protein is obtained in a couple of hours.

h) p. 7, second paragraph is particular confusing regarding what H₂O₂ concentrations are needed and whether the products are sulfonylated or sulfenylated. The authors need to first explain the chemical differences b/w these forms to call attention to the small difference in spelling. Then present results for one form and then the second form, rather than mixing the results b/w the two forms.

Authors: the whole section has been rewritten:

We next set out to identify the redox modifications of M^{Pro} by mass spectrometry-based redox proteomics and Western blot analysis (**Fig. 4**). This included the analysis of oxidized cysteine species (sulfenic acid (mono-oxidized) and sulfonic acid (tri-oxidized)), disulfide bridges and lysine-cysteine NOS/SONOS bridges. Determination of sulfenic acids via dimedone tagging confirmed that the cysteines in M^{Pro} exhibit in general a rather low sensitivity towards oxidation (**Fig. 4a**). Only treatment with supraphysiological concentrations of H₂O₂ (20 mM) leads to a substantial formation of sulfenic acids and indicates formation of irreversible overoxidation (sulfenic acid easily oxidizes to higher oxidation states). At H₂O₂ concentrations up to 1 mM, mass spectrometry in combination with dimedone tagging identified residues C145 (catalytic residue), C156, and C300 as becoming sulfenylated (**Fig. 4b, SI Fig. 17**). A H₂O₂ concentration-dependent sulfonylation was found for the catalytic cysteine C145 and, to a lesser extent, also for C117. Both cysteines form relatively small fractions of the irreversibly oxidized sulfonic acid up to 1 mM H₂O₂ in line with the reversible redox switching of M^{Pro} under these conditions (see above). Residues C85 and C300 are found to be oxidized to sulfonic acid particularly at supraphysiological concentrations (20 mM), whereas no H₂O₂ concentration-dependent oxidation of the other cysteines was obvious (**SI Fig. 18, 19**).

i) "variant C117S was the only variant" -- awkward. How about "C117S was the only variant"?

Authors: Changed accordingly.

j) p. 8: "the reduced state indicates that residues C145 and C117 cannot directly form a disulfide bond as residue N28 lies in between the two side chains" -- confusing. How about "cannot form without significantly structural changes"?

Authors: Newly written.

Inspection of the X-ray structure of M^{pro} determined in the reduced state indicates that residues C145 and C117 cannot directly form a disulfide bond as residue N28 **is bound** in between the two side chains (**Fig. 4c**).

k) p. 9: "Establishment of the second crosslink with K137 provided 12.0 kcal/mol" -- confusing. Maybe the authors mean that, with the second crosslink, the binding enthalpy increases to 12 kcal/mol?

Authors: Correct! Wording changed accordingly. Thank you.

Establishment of the second crosslink with K137 increases the binding enthalpy to 12 kcal/mol, ...

l) SI Fig. 26 caption: "MAH crosslinking is shown in Figure 4c of the main manuscript" -- they probably meant Fig. 6c?

Authors: Corrected accordingly.

With the revisions made and the novel data included we hope that our manuscript becomes acceptable for publication. We would like to thank the reviewers again for their thorough survey and constructive comments.

REVIEWER COMMENTS

Reviewer #1 (Remarks to the Author):

Overall, the authors have done good job of addressing the concerns of this Review. One point that is not fully addressed was: “In Figure 6: The authors show MAH causes monomer formation in WT but have no data on other Mpro’s. It would be very useful to see that C145S does not undergo this change to monomer thus implicating the requirement for C145. Also, data on the other mutants for the reactive residues in question could provide useful insight as well.” It would help if the authors could test MAH in the C145S mutant to show that it does not become monomer. Doing this would improve the manuscript.

In previous Comment 6, we raised the concern that the drugs seemed more active in culture than one would predict based on the protease activity, at least as compared to other reported agents. As now seen in SI Fig. 34, there is some cytotoxicity. We would suggest that the reviewers examine an agent that has some toxicity but is not predicted to have activity against MPro as a comparison. Barring that, they should be more open to the possibility that cytotoxicity is contributing to activity.

Reviewer #2 (Remarks to the Author):

We have now reviewed the second version of the manuscript by Dr. Tittmann entitled “Multiple redox switches of the SARS-CoV-2 main protease in vitro provide new opportunities for drug design”. We understand that this work is attractive thanks to a superb technical presentation, with an overwhelming number of figures and data to try to support the claims. Unfortunately, we are not convinced that these bridges have a regulatory role in Mpro activity. Most of our concerns have not been solved in the second version.

Reviewer #3 (Remarks to the Author):

The authors have addressed concerns from this reviewer. The revised manuscript is ready for acceptance.

Reviewer #4 (Remarks to the Author):

The authors have addressed the comments raised in the previous review. The presentation of the manuscript is much improved, and the discussion is also more comprehensive.

Reviewer #5 (Remarks to the Author):

This revision has addressed all my previous comments.

Response to reviewers

On behalf of all coauthors, I would like to thank all reviewers for the constructive comments and thorough survey of our manuscript

“Multiple redox switches of the SARS-CoV-2 main protease in vitro provide new opportunities for drug design”

submitted as an article to *Nature Communications*.

In light of the reviewers' comments, we have revised our manuscript and now include novel data as suggested by reviewer 1. **Changes are indicated in yellow shade**. A point-by-point discussion is given below.

Reviewer 1

1) Overall, the authors have done good job of addressing the concerns of this Review. One point that is not fully addressed was: “In Figure 6: The authors show MAH causes monomer formation in WT but have no data on other Mpro's. It would be very useful to see that C145S does not undergo this change to monomer thus implicating the requirement for C145. Also, data on the other mutants for the reactive residues in question could provide useful insight as well.” It would help if the authors could test MAH in the C145S mutant to show that it does not become monomer. Doing this would improve the manuscript.

Authors: We now include these data (novel SI Fig. 29) and have also added a small discussion in the main body of the ms.

Page 10:

Mass spectrometric analysis identified catalytic C145 and K137 **as the main** MAH-binding residues (**Fig. 7c, SI Fig. 28**). As both residues are separated by ~20 Å in all structures determined to date,

they could not simultaneously react with MAH without a structural change. Interestingly, the dimer interface, including residues E166 and N-terminal S1' from the second chain, is located right between C145 and K137 rationalizing why a crosslink between the two residues destabilizes the dimer (**Fig. 7d**). Also, we observed a substantial amount of MAH mono-links to interface residue C300, and, to a lesser extent, to C117. In line with these observations, MAH-induced monomerization is almost absent in variant C145S, whereas reaction of MAH with variants C117S and C300S leads to a complete (C117S) or strong monomerization (C300S) (**SI Fig. 29**).

2) In previous Comment 6, we raised the concern that the drugs seemed more active in culture than one would predict based on the protease activity, at least as compared to other reported agents. As now seen in SI Fig. 34, there is some cytotoxicity. We would suggest that the reviewers examine an agent that has some toxicity but is not predicted to have activity against MPro as a comparison. Barring that, they should be more open to the possibility that cytotoxicity is contributing to activity.

Authors: While we understand the impetus for asking for such a control, this becomes a bit ambiguous as this critically depends from the applied concentrations, incubation time etc. Also, it is not directly related to our main findings with respect to the redox switches in Mpro. We do agree with the reviewer that cytotoxicity is contributing to the antiviral activity and have changed both the text in the main ms as well as in the SI Figures 32 & 33.

Page 11:

Using a cell-based SARS-CoV-2 infection model we can demonstrate as proof-of-concept that both MAH as well as BMOE principally exhibit antiviral activity (more than 50-fold reduced virus progeny) with only a mild cytotoxic effect (**SI Fig. 32, SI Fig. 33**). Quantitative analysis of the dose response curves (virus RNA progeny) shows that the estimated EC₅₀ values are in the millimolar range and thus higher than the IC₅₀ values obtained under in vitro conditions with highly enriched M^{Pro} in buffer (**SI Fig. 34**). The EC₅₀ value for BMOE is ~10 times smaller (0.8 mM) than that of MAH (7.9 mM) akin to the in vitro studies and estimated IC₅₀ values. The almost identical ratios for EC₅₀ (EC₅₀^{BMOE}/EC₅₀^{MAH}) and IC₅₀ values (IC₅₀^{BMOE}/IC₅₀^{MAH}) would seemingly suggest that antiviral activity of these compounds is reflecting inhibition of M^{Pro}. Admittedly, we cannot rule out that these compounds inhibit viral replication by “off-target” effects on other viral proteins, e.g. the papain-like protease, or host proteins. Our systematic analysis of MAH and BMOE cytotoxicity implies that the antiviral activity of both compounds mostly reflects inhibition of M^{Pro} but cytotoxicity

is also contributing to a smaller extent (SI Fig. 35). Both compounds exhibit a mild cytotoxic effect in the concentration regime relevant for the dose response analysis, in particular BMOE, but become clearly cytotoxic at higher concentrations. We would like to note that cytotoxicity of both compounds might be more pronounced in other cell lines.

Legend to SI Fig. 32

ED Figure 32. MAH inhibits SARS-CoV-2 propagation and the synthesis of viral proteins with mild cytotoxicity. (a) Reduced cytopathic effect (CPE) upon treatment with MAH. Vero E6 cells were treated with 20 mM MAH or the PBS control for 1 h before infection, and then throughout the time of infection (48 h). Cell morphology was assessed by bright field microscopy. Note that the CPE was clearly visible in virus-infected cells but to a far lesser extent upon treatment with MAH. (b) Cytotoxicity by MAH. Vero E6 cells were treated with 20 mM MAH for 48 h. The release of lactate dehydrogenase (LDH) to the supernatant was quantified by bioluminescence as a read-out for cytotoxicity. The percentages reflect the proportion of LDH released to the media, compared to the overall amount of LDH in the cells (LDH control) (mean with SD, n=3). A systematic analysis of MAH cytotoxicity is provided in SI Fig. 35.

Legend to SI Fig. 33

SI Figure 33. BMOE inhibits SARS-CoV-2 propagation and the synthesis of viral proteins with mild cytotoxicity. (a) Reduced cytopathic effect (CPE) upon treatment with BMOE. Vero E6 cells were treated with 20 mM BMOE or the PBS control for 1 h before infection, and then throughout the time of infection (48 h). Cell morphology was assessed by bright field microscopy. Note that the CPE was clearly visible in virus-infected cells but to a far lesser extent upon treatment with BMOE. (b) Cytotoxicity by BMOE. Vero E6 cells were treated with 20 mM BMOE for 48 h. The release of lactate dehydrogenase (LDH) to the supernatant was quantified by bioluminescence as a read-out for cytotoxicity. The percentages reflect the proportion of LDH released to the media, compared to the overall amount of LDH in the cells (LDH control) (mean with SD, n=3). A systematic analysis of BMOE cytotoxicity is provided in SI Fig. 35.

With the revisions made and the novel data included we hope that our manuscript becomes acceptable for publication. We would like to thank the reviewers again for their thorough survey and constructive comments.

REVIEWERS' COMMENTS

Reviewer #1 (Remarks to the Author):

The author have addressed our concerns and we recommend acceptance.